# Neural Symplectic Form: Learning Hamiltonian Equations on General Coordinate Systems

**Yuhan Chen**
Kobe University
Kobe, Japan 657–8501
193x226x@stu.kobe-u.ac.jp

**Takashi Matsubara**
Osaka University
Osaka, Japan 560–8531
matsubara@sys.es.osaka-u.ac.jp

**Takaharu Yaguchi**
Kobe University
Kobe, Japan 657–8501
yaguchi@pearl.kobe-u.ac.jp

## Abstract

In recent years, substantial research on the methods for learning Hamiltonian equations has been conducted. Although these approaches are very promising, the commonly used representation of the Hamilton equation uses the generalized momenta, which are generally unknown. Therefore, the training data must be represented in this unknown coordinate system, and this causes difficulty in applying the model to real data. Meanwhile, Hamiltonian equations also have a coordinate-free expression that is expressed by using the symplectic 2-form. In this paper, we propose a model that learns the symplectic form from data using neural networks, thereby providing a method for learning Hamiltonian equations from data represented in general coordinate systems, which are not limited to the generalized coordinates and the generalized momenta. Consequently, the proposed method is capable not only of modeling the target equations of both Hamiltonian and Lagrangian formalisms but also of extracting unknown Hamiltonian structures hidden in the data. For example, many polynomial ordinary differential equations such as the Lotka–Volterra equation are known to admit non-trivial Hamiltonian structures, and our numerical experiments show that such structures can certainly be learned from data. Technically, each symplectic 2-form is associated with a skew-symmetric matrix, but not all skew-symmetric matrices define a symplectic 2-form. In the proposed method, using the fact that symplectic 2-forms are derived as the exterior derivative of certain differential 1-forms, we model the differential 1-form by neural networks, thereby improving the efficiency of learning.

## 1   Introduction

In recent years, the applications of deep learning to learn the fundamental equations of classical mechanics have been actively studied. Analytical mechanics, which is a theory of classical mechanics, is classified into Lagrangian mechanics and Hamiltonian mechanics [1, 3, 18]. The equations of motion of Lagrangian mechanics are called the Euler–Lagrange equation, for which several neural network models were proposed [7, 17]. In Lagrangian mechanics, the equations are described using the state variables and the time derivatives of them. This feature of Lagrangian mechanics makes it easy to prepare the data necessary for learning. On the other hand, Hamiltonian mechanics can describe more general equations which are not covered by Lagrangian mechanics. In the previous

35th Conference on Neural Information Processing Systems (NeurIPS 2021).

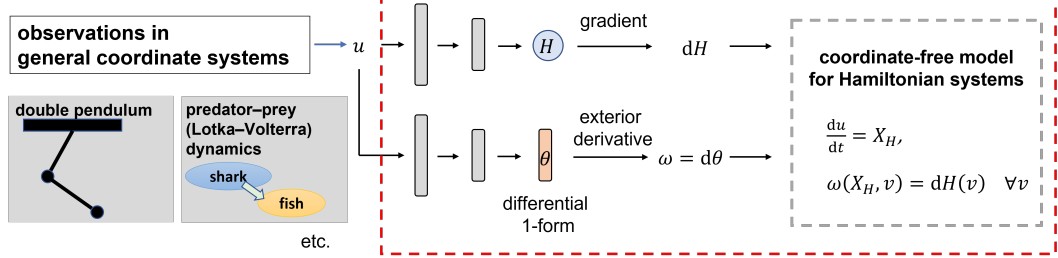

Figure 1: Overview of the proposed method. Generally, the analytical representation of generalized momenta is unknown, so the data cannot be presented in a canonical coordinate system. The proposed method learns the Hamilton equation from data represented in an arbitrary coordinate system by learning the symplectic 2-form as well as the energy function. In particular, to ensure that the learned symplectic 2-form is closed, our method learns the differential 1-form that derives the symplectic 2-form. A universal approximation theorem is also provided.

models for Hamiltonian equations, the equation

$$\frac{\mathrm{d}}{\mathrm{d}t}\begin{pmatrix} q \\ p \end{pmatrix} = \begin{pmatrix} O & I \\ -I & O \end{pmatrix} \begin{pmatrix} \frac{\partial H}{\partial q} \\ \frac{\partial H}{\partial p} \end{pmatrix} \tag{1}$$

is typically assumed, where $q$ is the state variable, and $p$ is a variable called the generalized momentum. It is known that Hamiltonian equations can be written in this form by using these variables; more precisely, any Hamiltonian equation can be locally written as (1) by using a special coordinate system called the Darboux coordinate system [20]. However, this coordinate system depends on the generally unknown Hamiltonian (the energy function), and it is usually not possible to prepare data in this coordinate system.

On the other hand, Hamiltonian equations also have a coordinate-free representation using the symplectic 2-form [20]. In this paper, we propose a method to learn the symplectic 2-form from data by using neural networks, thereby introducing a method to learn the Hamilton equation from data represented in general coordinate systems, not restricted to the generalized momentum. As explained below, general 2-forms are corresponding to skew symmetric matrices. Hence, a naive way to learn the symplectic 2-form is learning a skew symmetric matrix; however, actually, the equation learned in this way may not be Hamiltonian because not all skew symmetric matrices are corresponding to symplectic 2-forms. In the proposed method, instead of learning the 2-form directly, a 1-form that derives the symplectic 2-form is learned (see Figure 1).

Because the proposed model can be applied to data in any coordinate system, it can be employed for extracting the unknown underlying Hamiltonian structure behind the data that are expected to have such a structure. For example, many polynomial ordinary differential equations are known to have a Hamiltonian structure, but it is not easy to find the structure analytically. The proposed method extracts the hidden structure in such cases from data. If the Hamiltonian structure can be found, it is possible to make predictions while preserving the energy conservation law. In addition to the energy, other hidden conserved quantities can be also extracted.

Main contributions of this paper include:

**Symplectic geometric approach to learning symplectic 2-forms.** The symplectic 2-form required to describe the Hamilton equation corresponds to a skew-symmetric matrix, but conversely, not all skew-symmetric matrices correspond to a symplectic 2-form. In this paper, we propose an efficient model by learning the symplectic 1-form that derives the symplectic 2-form with neural networks.

**Learning the Hamilton equation from data in arbitrary coordinate systems.** By using the coordinate-free representation of the Hamilton equation, it is possible to learn the Hamilton equation from data represented in a general coordinate system, not restricted to the Darboux coordinate system. In this way, the proposed method can determine whether the given data can be explained by the hidden theory of classical mechanics or not. A universal approximation theorem is also provided.

Table 1: Comparison with other studies.

| | HNN [11] | LNN [7] | Skew Matrix Learning (Sec. 3) | Neural Symplectic Form (proposed) |
|---|---|---|---|---|
| In the known Darboux coordinate | yes | yes | yes | yes |
| In general coordinates on cotangent bundles | | yes | yes | yes |
| On general symplectic manifolds | | | yes | yes |
| Only symplectic forms | N/A | N/A | | yes |

## 2 Related Work

**Neural Networks for Hamiltonian Mechanics**  Neural ordinary differential equation (NODE, [5]) is a neural network that models the time-derivative of the states, thereby defining an ordinary differential equation (ODE) in a general way. Due to the generality, this model does not admit the energy conservation law.

Hamiltonian neural network (HNN) is a neural network that models the Hamiltonian $H$ and defines the dynamics following the Hamiltonian mechanics, thereby ensuring the energy conservation law [11]. Although models of the form (1) are often used in the previous studies, the Hamilton equation has this form only in the Darboux coordinates [20]. The Darboux coordinate system, which is essentially the generalized momentum $p$, is defined using the Hamiltonian $H$, which is the target to be learned. Hence, training data in the coordinate is usually unavailable. In addition, Hamiltonian equations are defined on general symplectic manifolds; however, in the existing studies, cotangent bundles are typically assumed as the symplectic manifold.

A numerical integration of the Hamiltonian system is known to destroy the symplectic structure and does not conserve the Hamiltonian $H$, unless the integrator is carefully designed (see, e.g., [12, 26]). Several studies focus on the numerical integration that conserves the Hamiltonian $H$ [6, 9, 19, 27, 28]. The HNN was extended to energy-conserving partial differential equation (PDE) systems, such as the Korteweg–De Vries (KdV) equation [19], to dissipative systems, such as a pendulum with friction [19, 29]. In [15], a discrete-time model is proposed for Poisson systems, which are extension of Hamiltonian systems, where the skew symmetric matrix can be degenerate. In particular, in [15], dynamics with state-dependent skew symmetric matrices are leaned by introducing coordinate transformations for learning the dynamics in the latent space. The proposed method is different from this study in that our method does not use the coordinate transformations and hence has an advantage in interpretability. Another approach is employed in [4, 16], where the symplectic map is modeled. SympNets [16] are also shown to be universal approximators, and, in [4], a bound on the prediction error is provided. The proposed method can be combined with these discrete-time approaches.

Modeling using machine learning has also been performed in the field of quantum mechanics, for example, by Tkatchenko and coworkers (e.g. [22, 25]). Some breakthroughs have been reported that have not been possible with conventional computational chemistry methods. The relationship with these studies needs to be investigated in the future.

**Neural Networks for Lagrangian Mechanics**  Another branch of studies focus on Lagrangian mechanics. Lagrangian neural network (LNN) is a neural network that models the Lagrangian $L$ in a general way [7], and deep Lagrangian network explicitly defines the kinetics energy with a trainable mass matrix [17]. Lagrangian mechanics defines Lagrangian systems on tangent bundles [18], where the state is the pair of the position $q$ and velocity $\dot{q}$. The systems have specific symplectic structures, which are equivalent to Hamiltonian systems in general coordinate systems on cotangent bundles. Because the LNN does not assume equations of a specific form, it can learn a wider class of systems, including a double pendulum, in addition to the systems that the HNN can learn (see Table 1). Similarly to the HNN, numerical integrators that preserve the symplectic structure have been investigated [8, 23]. Neural network architectures that ensure translational and rotational symmetries have also been investigated [10, 24].

As mentioned before, not all Hamiltonian systems are defined on cotangent bundles. For examples, the some polynomial equations including the Lotka–Volterra equation have a different symplectic structure [13]. In fact, the Lotka–Volterra equation is actually a Hamiltonian system, even though its

states are not position, velocity, nor generalized momentum. These equations are out of the scopes of the HNN and LNN (see Table 1).

## 3 Methods

**Coordinate-Free Representation of Hamiltonian Equations**  In this paper, we propose a neural network model based on a coordinate-free representation of Hamiltonian equations. First, we describe this representation. Because detailed knowledge of geometry is required for a precise description of this representation, the details are omitted here. For more details, see Appendix A.

The model in this paper can be used on a general symplectic manifold, but for simplicity, we describe the case where the phase space is $\mathcal{M} = \mathbb{R}^{2N}$. A differential 2-form $\omega$ on $\mathcal{M}$ is a skew-symmetric bilinear function that maps given two vectors into a real number, depending on each point $u$ on $\mathcal{M}$. The skew-symmetric bilinear function defined by $\omega$ has the following matrix representation:

$$\omega_u(v_1, v_2) = v_1^\top W_u v_2, \quad \text{for all } v_1, v_2 \in \mathbb{R}^{2N},$$

where $W_u$ is a skew-symmetric matrix, and the subscript $u$ denotes that $\omega$ and hence its matrix representation $W_u$ depend on $u$. A symplectic 2-form is a differential 2-form that is nondegenerate and closed. See Appendix A for the precise meanings of these terms.

The following is the coordinate-free form of Hamiltonian equations [18]

$$\frac{\mathrm{d}u}{\mathrm{d}t} = X_H, \ \omega(X_H, v) = \mathrm{d}H(v) \quad \text{for all } v \in \mathbb{R}^{2N}. \tag{2}$$

Here, $\mathrm{d}H$ is the Fréchet derivative of the Hamiltonian $H$, and $X_H$ is a vector field depending on $H$. This equation is satisfied regardless of the coordinate system in which the state variable $u$ is expressed. Therefore, by using this equation as a model, as long as the given data is described by the Hamilton equation, it is possible to learn both the symplectic 2-form and the Hamiltonian that define the Hamilton equation, no matter what coordinate system the data is given in.

**Naive method: the skew matrix learning**  By replacing the symplectic 2-form with the matrix $W_u$, (4) can be rewritten as

$$\frac{\mathrm{d}u}{\mathrm{d}t} = X_H, \ X_H^\top W_u v = v \cdot \nabla H \text{ for all } v \in \mathbb{R}^{2N} \quad \Leftrightarrow \quad \frac{\mathrm{d}u}{\mathrm{d}t} = W_u^{-\top} \nabla H.$$

A natural model based on this representation would be a model in which $W_u$ and $H$ are modeled by multilayer perceptrons:

$$\frac{\mathrm{d}u}{\mathrm{d}t} = W_{u,\mathrm{NN}}^{-\top} \nabla H_{\mathrm{NN}}(u), \tag{3}$$

where $H_{\mathrm{NN}}$ is a function of $u$ defined by a multilayer perceptron, and $W_{u,\mathrm{NN}}$ is a skew-symmetric matrix depending on $u$ represented by another multilayer perceptron. We refer this model as *the skew matrix learning*. If $u$ is represented in the Darboux coordinate system, (3) becomes (1), and hence the model is the same as the Hamiltonian neural network.

Similarly to Hamiltonian neural networks, this model has the energy conservation law.

**Theorem 1.** *Solutions to* (3) *satisfy* $\mathrm{d}H_{\mathrm{NN}}/\mathrm{d}t = 0$.

See Appendix C for the proof.

**Neural Symplectic Form**  Although the model in the previous section is natural, it is a redundant model and inefficient for learning as explained below. As explained in Introduction, although a differential 2-form corresponds to a skew-symmetric matrix, not all skew-symmetric matrices define a symplectic 2-form. Symplectic 2-forms have the characteristic feature of being closed. In this paper, we propose a model in which the learned 2-form is guaranteed to be closed. We refer the learned 2-form as *the neural symplectic form*.

First, the necessary terminology is briefly explained. See Appendix A for details. A differential 0-form on $\mathcal{M} = \mathbb{R}^{2N}$ is a function from $\mathcal{M}$ to $\mathbb{R}$. A differential 1-form $\theta$ on $\mathcal{M} = \mathbb{R}^{2N}$ is a field of linear functions each of which maps a vector $v \in \mathbb{R}^{2N}$ to $\theta_u(v) \in \mathbb{R}$, depending on each point

$u \in \mathcal{M}$. In general, a linear function from $\mathbb{R}^{2N}$ to $\mathbb{R}$ can be expressed as an inner product with a vector, so a differential 1-form can be expressed as a vector field depending on $u$. A differential operation called the exterior derivative d is defined for differential forms. The exterior derivative is a graded linear map, i.e. a linear map depending on an integer $k$, which transfers a differential $k$-form to a differential $(k+1)$-form and has the property that $\mathrm{dd} = 0$.

The differential form in Ker d is called a closed form. Since a symplectic 2-form is a closed form, in order to learn the symplectic 2-form by neural networks, neural networks should be designed so that differential 2-forms represented by the neural networks are contained in Ker d. Meanwhile, due to the property of the exterior derivative, $\mathrm{dd} = 0$, it holds that Im d $\subset$ Ker d.

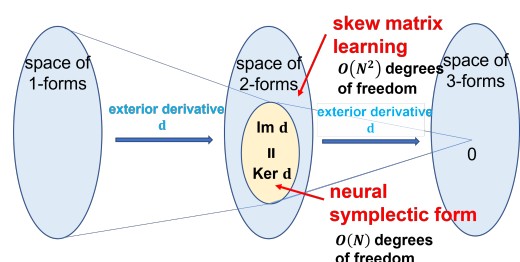

Actually, according to the de Rham theorem, when the phase space is $\mathbb{R}^{2N}$, these two spaces coincide: Im d $=$ Ker d. The difference between these two spaces Im d/Ker d is called the cohomology space. The de Rham theorem states that the cohomology space is isomorphic

Figure 2: The de Rham theorem ensures the Ker d $=$ Im d.

to the homology space, which is roughly a space of "spatial holes." Because $\mathbb{R}^{2N}$ contains no holes, the homology space must vanish, and hence Im d $=$ Ker d holds (see Figure 2.) Even when the phase space has a hole, in many cases the space can be embedded in a large Euclid space without holes, and the model should be defined on that space. See Appendix B.

Therefore, in this paper, instead of learning the symplectic 2-form directly, we propose a method to learn the symplectic 2-form by learning the differential 1-form of which exterior derivative gives the symplectic 2-form. The following is the coordinate-free form of the proposed model:

$$\tilde{\omega} = \mathrm{d}\theta_{\mathrm{NN}}, \quad \frac{\mathrm{d}u}{\mathrm{d}t} = \tilde{X}_{H_{\mathrm{NN}}}, \quad \omega(\tilde{X}_{H_{\mathrm{NN}}}, v) = \mathrm{d}H_{\mathrm{NN}}(v) \quad \text{for all } v \in \mathbb{R}^{2N} \tag{4}$$

Unlike (3), in this model skew-symmetric matrices that do not correspond to symplectic forms are not explored. Therefore, this model can be trained much more efficiently than (3). In fact, suppose that the dimension of the phase space is $2N$. Then, the number of components of the skew-symmetric matrix is $N(2N - 1)$, while the differential 1-form is represented as a vector field, the number of which components is $2N$. Consequently, a model using the neural symplectic form can reduce the order of the number of functions to be learned from $O(N^2)$ to $O(N)$ without sacrificing the expressive power.

As per the above, the differential 1-form can be expressed as a vector field. Hence the neural network for modeling the 1-form in the proposed model essentially models a vector field $Y_{\mathrm{NN}}$, which represents the differential 1-form $\theta_{\mathrm{NN}}$. As shown in Appendix A, the vector field $Y_{\mathrm{NN}}$ is transformed to the matrix $\tilde{W}_u$ representing the symplectic 2-form as follows:

$$(\tilde{W}_u)_{i,j} = \frac{\partial(Y_{\mathrm{NN}})_i}{\partial u_j} - \frac{\partial(Y_{\mathrm{NN}})_j}{\partial u_i}.$$

In the actual model, $Y_{\mathrm{NN}}$ given by the neural network is differentiated by the automatic differentiation, and $W_{u,\mathrm{NN}}$ is obtained by substituting the derivatives into the above equation. Thus we have the model expressed in terms of vectors and a matrix, without using the differential forms:

$$\frac{\mathrm{d}u}{\mathrm{d}t} = \tilde{W}_u^{-\top}\nabla H_{\mathrm{NN}}(u),$$

$$(\tilde{W}_u)_{i,j} = \frac{\partial(Y_{\mathrm{NN}})_i}{\partial u_j} - \frac{\partial(Y_{\mathrm{NN}})_j}{\partial u_i}. \tag{5}$$

The proposed model has the energy conservation law and the universal approximation property. For the proofs, see Appendix C.

**Theorem 2.** *Solutions to (4), or equivalently, to (5) satisfy $\mathrm{d}H_{\mathrm{NN}}/\mathrm{d}t = 0$.*

**Theorem 3.** *Suppose that the Hamilton equation to be learned can be represented in the form (2) using a symplectic 2-form $\omega$ derived from a 1-form $\theta \in W^{\mathrm{d},p}$ and the Hamiltonian $H \in W^{1,p}$.*

*Suppose also that the phase space is compact. The model* (4) *with multilayer perceptrons with a sufficiently smooth activation function has the universal approximation property in the sense that $\theta$ and $H$ can be approximated by the multilayer perceptrons with arbitrary accuracy.*

$W^{1,p}$ and $W^{\mathrm{d},p}$ are Sobolev spaces; see [2, 14].

**Learning dynamics by using the neural symplectic form** The proposed model learns the symplectic form and the Hamiltonian by minimizing the squared error between the left- and right-hand sides of (4), assuming that time series data of state vectors $\{u^{(n)}\}$ and its time-derivatives $\{\frac{\mathrm{d}u}{\mathrm{d}t}^{(n)}\}$ are given;

$$\text{minimize} \quad \sum_n \left| \frac{\mathrm{d}u}{\mathrm{d}t}^{(n)} - \tilde{W}_{u^{(n)}}^{-\top} \nabla H_{\mathrm{NN}}(u^{(n)}) \right|^2, \quad (\tilde{W}_{u^{(n)}})_{i,j} = \frac{\partial (Y_{\mathrm{NN}})_i}{\partial u_j}(u^{(n)}) - \frac{\partial (Y_{\mathrm{NN}})_j}{\partial u_i}(u^{(n)}).$$

If the time-derivatives are not available, interpolated data should be used. For each state vector $u^{(n)}$, first $Y_{\mathrm{NN}}(u^{(n)})$ is computed. Then, the derivatives of $Y_{\mathrm{NN}}(u^{(n)})$ are calculated by the automatic differentiation to obtain $(\tilde{W}_u^{-\top})_{i,j}$. Meanwhile, the gradient of the Hamiltonian is computed in the same way; first $H_{\mathrm{NN}}(^{(n)})$ is computed, and then $\nabla H_{\mathrm{NN}}(^{(n)})$ is obtained by the automatic differentiation.

In the standard Hamiltonian neural networks only the computation of the gradient of the Hamiltonian is required. Hence the computational cost of the neural symplectic form is roughly $2N$ times the computational cost of Hamiltonian neural networks, where $2N$ is the number of the state variables, because the number of components of $Y_{\mathrm{NN}}(u^{(n)})$ is $2N$. When $N$ is large, such as in many-body problems, the most computationally expensive part may be computation of the inverse of the matrix $\tilde{W}_{u^{(n)}}$, which requires $O(N^3)$ operations. To avoid the computation of the inverse, one may be tempted to minimize

$$\sum_n \left| \tilde{W}_{u^{(n)}}^{\top} \frac{\mathrm{d}u}{\mathrm{d}t}^{(n)} - \nabla H_{\mathrm{NN}}(u^{(n)}) \right|^2.$$

However, this model does not work because this is trivially minimized by setting $\tilde{W}_{u^{(n)}}^{\top} = O$ and $H_{\mathrm{NN}} = (\text{constant.})$ Learning the inverse while maintaining the symplectic structure is future work.

## 4 Numerical Examples

**Outline of experiments** The proposed method with the neural symplectic form can model general Hamiltonian equations on general symplectic manifolds using data represented by general coordinate systems. We illustrate these advantages in the following numerical experiments with comparative methods. The details of the data in the numerical experiments along with additional results are provided in Appendix D.

We performed the experiments using HNN, LNN[1], skew matrix learning and neural symplectic form. The energy function, the skew-symmetric matrix, and the 1-form for the neural symplectic 2-form were modeled by using a neural network that had two hidden layers of 200 units and the $\tanh$ activation function. We used 80 percent of collected data for training and the remaining for test. We trained each model 10 times using the Adam optimizer with a learning rate of $10^{-3}$ for 2000 iterations. Only for LNN, we set the learning rate to $10^{-4}$ due to an instability of learning; in our experiments the loss function of LNN sometimes did not converge monotonically, but oscillated when the learning rate was set to $10^{-3}$. Hence we truncated the training process at the iteration where the loss function achieved the best score. This oscillation should be due to the non-uniqueness of Lagrangian; see Appendix D for details.

The experiments were performed on NVIDIA A100 with double precision. After training, we evaluated the models using the squared time-derivative errors of the test subset. Using SciPy odeint under the default setting, we predicted 10 orbits from random initial values and obtained the errors in the system energy.

---

[1]We used the implementation of the LNN model in torchdyn [21] https://github.com/DiffEqML/torchdyn/blob/master/docs/tutorials/09_lagrangian_nets.ipynb (Apache 2.0 License)

**Mass-Spring System**   First, we investigate a Hamiltonian system in a general coordinate on a cotangent bundle, namely, a simple mass-spring system, depicted in Figure 3. The equation of motion of this system is

$$\frac{\mathrm{d}}{\mathrm{d}t}\begin{pmatrix} q_1 \\ q_2 \\ v_1 \\ v_2 \end{pmatrix} = \begin{pmatrix} v_1 \\ v_2 \\ -\frac{k_1}{m_1}(q_1 - l_1) + \frac{k_2}{m_1}(q_2 - q_1 - l_2) \\ -\frac{k_2}{m_2}(q_2 - q_1 - l_2) \end{pmatrix},$$

Figure 3: The mass-spring system.

which is a Hamiltonian system with the energy function

$$H(q_1, q_2, p_1, p_2) = \frac{p_1^2}{2m_1} + \frac{p_2^2}{2m_2} + \frac{k_1(q_1 - l_1)^2}{2} + \frac{k_2(q_2 - q_1 - l_2)^2}{2},$$

where $q_1$, $q_2$ are the positions of the mass points, and $p_1$, $p_2$ are the momenta, which are defined by $p_1 = m_1 v_1, p_2 = m_2 v_2, v_1 = \mathrm{d}q_1/\mathrm{d}t, v_2 = \mathrm{d}q_2/\mathrm{d}t$. Suppose that the exact values of $m_1$ and $m_2$ are unknown, and only the positions $q_1$ and $q_2$ and their derivatives are given. Although $m_1$ and $m_2$ may be estimated from the given states, for evaluation of the models, we tried to learn the dynamics from the given states directly.

The time-derivative errors and the energy errors are shown in Tables 2 and 3, respectively. As explained in Appendix D, if the state variables are represented by the positions and the velocities, the equation of motion of this system cannot be written in the standard form. Nevertheless, as this is a simple problem, all the models work well. Most of the predicted orbits shown in Figure 4 have a good fit to the true one. Note that the errors shown in Table 2 are small enough. They are multiplied by $10^3$ and shown in the physical scale; i.e., the errors are not measured by using the normalized state variables but by using the state variables with the true scale. In the normalized scale, the error by LNN, for example, was 0.0095. This is also the case in the other experiments. Among the methods, NODE and the neural symplectic form performed better. The test time-derivative error of the skew matrix learning is a little larger. In terms of the energy error, the neural symplectic form and HNN are superior; however there is not much difference.

**Double Pendulum**   Next, we consider another Hamiltonian system, namely, a double pendulum shown in Figure 5, of which equation of motion is

$$\frac{\mathrm{d}\theta_1}{\mathrm{d}t} = \phi_1, \quad \frac{\mathrm{d}\theta_2}{\mathrm{d}t} = \phi_2,$$

$$\frac{\mathrm{d}\phi_1}{\mathrm{d}t} = \frac{g(\sin\theta_2 \sin(\theta_1 - \theta_2) - \frac{m_1 + m_2}{m_2}\sin\theta_1) - (l_1\theta_1^2 \cos(\theta_1 - \theta_2) + l_2\theta_2^2)\sin(\theta_1 - \theta_2)}{l_1(\frac{m_1 + m_2}{m_2} - \cos^2(\theta_1 - \theta_2))},$$

$$\frac{\mathrm{d}\phi_2}{\mathrm{d}t} = \frac{\frac{g(m_1 + m_2)}{m_2}(\sin\theta_1 \cos(\theta_1 - \theta_2) - \sin\theta_2) - (\frac{l_1(m_1 + m_2)}{m_2}\theta_1^2 + l_2\theta_2^2 \cos(\theta_1 - \theta_2))\sin(\theta_1 - \theta_2)}{l_2(\frac{m_1 + m_2}{m_2} - \cos^2(\theta_1 - \theta_2))}.$$

(6)

For the energy function and the Lagrangian of this system, see Appendix D. The generalized momenta of this system are

$$p_1 = (m_1 + m_2)l_1^2\phi_1 + m_2 l_1 l_2\phi_2 \cos(\theta_1 - \theta_2), \quad p_2 = m_2 l_2^2\phi_2 + m_2 l_1 l_2\phi_1 \cos(\theta_1 - \theta_2).$$

Because the generalized momenta are not obvious, we assume that the data of $\theta_1, \theta_2, \phi_1, \phi_2$ are given instead of $\theta_1, \theta_2, p_1, p_2$, where $\phi_1, \phi_2$ are the time derivatives of $\theta_1, \theta_2$.

The test time-derivative errors are shown in Table 2. Firstly, (6) cannot be written as the standard form (1) with a certain Hamiltonian. Hence, the test time-derivative error of HNN cannot be completely zero when learned using states $\theta_1(t), \phi_1(t), \theta_2(t), \phi_2(t)$. The result shown in Table 2 confirms this. The test time-derivative errors by NODE, the skew matrix learning and the neural symplectic form are much smaller than that by HNN. In particular, the error by the skew matrix learning is larger than that by the neural symplectic form. This is due to the non-existence of the one-to-one correspondence between the skew matrix and the symplectic 2-form, which decreases the efficiency of learning. NODE performed very well too; however, as explained below, this model failed to predict long-term behaviors.

Table 2: Test time-derivative errors.

| | NODE | HNN | LNN | Skew Matrix Learning | Neural Symplectic Form (proposed) |
|---|---|---|---|---|---|
| mass-spring | **0.17 ± 0.14** | 694.77 ± 26.37 | 882.52 ± 1753.93 | 102.07 ± 68.00 | 0.52 ± 0.71 |
| double pendulum | 8.65 ± 1.38 | 15.69 ± 0.44 | 269.52 ± 136.85 | 8.47 ± 1.22 | **4.02 ± 2.08** |
| Lotka–Volterra | 2.02 ± 5.55 | 227.03 ± 2.31 | N/A | 1.05 ± 0.76 | **0.46 ± 0.30** |

The best and second best results are emphasized by bold and underlined fonts, respectively. Multiplied by $10^3$ for the mass-spring system and the Lotka–Volterra equation. The experiments were conducted 10 times each. The results of LNN for double pendulum were computed using nine times except for a failed one.

Table 3: Energy errors.

| | NODE | HNN | LNN | Skew Matrix Learning | Neural Symplectic Form (proposed) |
|---|---|---|---|---|---|
| mass-spring | 0.840 ± 0.328 | 0.551 ± 0.112 | 2.281 ± 0.004 | 6.203 ± 7.555 | **0.368 ± 0.055** |
| double pendulum (T=5) | **1.070 ± 0.694** | (0.755 ± 0.320) | 17.740 ± 10.804 | 3.931 ± 7.266 | 6.400 ± 0.971 |
| double pendulum (T=30) | 11.240 ± 12.297 | N/A | N/A | 622982.133 ± 1814794.079 | **7.300 ± 3.925** |
| Lotka–Volterra | 0.578 ± 0.558 | 0.444 ± 0.458 | N/A | 0.041 ± 0.072 | **0.012 ± 0.013** |

The best and second best results of the true energies are emphasized by bold and underlined fonts, respectively. The differences of the true energy functions at $t = T$ and $t = 0$, where $T = 30$ for the Lotka–Volterra equation and $T = 5$ for the others. 10 orbits were simulated using randomly generated initial values. In the double pendulum test, the energy error of the HNN was best but this is because of the very small amplitudes of the predicted solutions; hence the result of NODE is cosidered best.

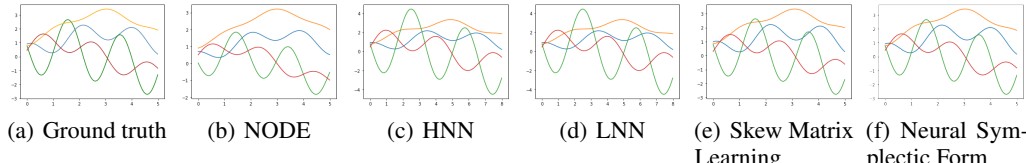

| (a) Ground truth | (b) NODE | (c) HNN | (d) LNN | (e) Skew Matrix Learning | (f) Neural Symplectic Form |

Figure 4: Example of the orbits predicted by the trained models for the mass-spring test. The horizontal axis represents time. Each component of $u(t) = (q_1(t), v_1(t), q_2(t), v_2(t))$ is represented: blue ($q_1$), green ($v_1$), orange ($q_2$), and red ($v_2$).

Examples of the predicted orbits are shown in Figure 6. As expected, HNN failed to predict the orbits. In the result by the skew matrix learning, although the speed of oscillation appears to be correct, the heights of the peaks are different from the true trajectory. Meanwhile, the prediction by the neural symplectic form achieved a better agreement with the true one than other models.

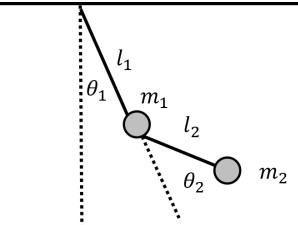

Figure 5: A double pendulum and the related constants and the variables.

The energy errors are shown in Table 3. The errors of NODE and the skew matrix learning are small for the short term prediction. However, Figure 7 shows an example of the long-term prediction results of these models and the proposed neural symplectic form. The results of the two NODEs are obtained using the models trained with different seeds. Since NODE does not have a Hamiltonian structure, the results gradually decay or diverge. The prediction results by the skew matrix learning also show a gradual increase in the state variables, and the results collapse after exceeding a certain value. On the other hand, the result by the neural symplectic form oscillates stably, although the range of oscillation is a little larger than the true orbit.

**Lotka–Volterra equation**   Systems of differential equations with a polynomial right-hand side are often Hamiltonian with a hidden symplectic structure. For example, the Hamiltonian structure of the

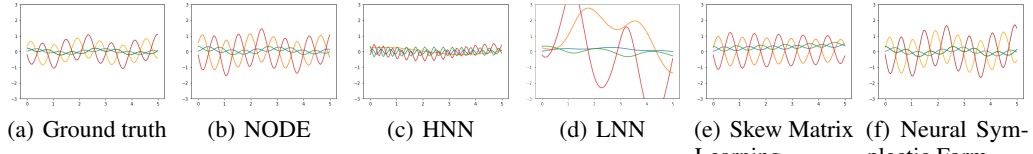

(a) Ground truth    (b) NODE    (c) HNN    (d) LNN    (e) Skew Matrix Learning    (f) Neural Symplectic Form

Figure 6: Example of the orbits predicted by the trained models for the double pendulum test. The horizontal axis represents time. Each component of $u(t) = (\theta_1(t), \phi_1(t), \theta_2(t), \phi_2(t))$ is represented: blue ($\theta_1$), orange ($\phi_1$), green ($\theta_2$), and red ($\phi_2$).

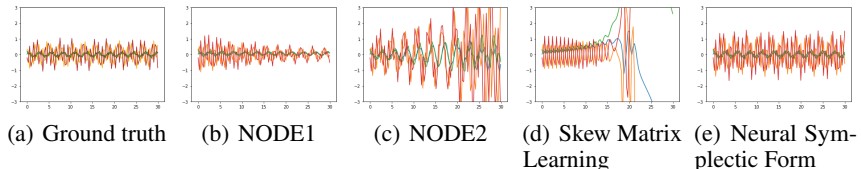

(a) Ground truth    (b) NODE1    (c) NODE2    (d) Skew Matrix Learning    (e) Neural Symplectic Form

Figure 7: Example of the orbits during a long period ($t = 30$) predicted by the trained models for the double pendulum test. NODE1 and NODE2 are results of two NODE models trained with a different random seed. The horizontal axis represents time. Each component of $u(t) = (\theta_1(t), \phi_1(t), \theta_2(t), \phi_2(t))$ is represented: blue ($\theta_1$), orange ($\phi_1$), green ($\theta_2$), and red ($\phi_2$).

generalized Lotka–Volterra equation

$$\frac{\mathrm{d}x_i}{\mathrm{d}t} = x_i(\sum_{j=1}^{m} a_{ij}\Pi_{k=1}^{l} x_k^{b_{jk}} + \lambda_i), \tag{7}$$

is investigated in Hernández-Bermejo and Fairén [13]. The right-hand side of this equation is quite general because this equation has the form of $x_i$(polynomial of the other state variables). For example, many mathematical compartment models in biology are of this form. In fact, the original Lotka–Volterra model is proposed as a model of a predator-prey dynamics.

We used a standard Lotka–Volterra model, which is included in (7), of the following form:

$$\frac{\mathrm{d}x_1}{\mathrm{d}t} = a_{12}x_1x_2 + \lambda_1 x_1, \quad \frac{\mathrm{d}x_2}{\mathrm{d}t} = a_{21}x_1x_2 + \lambda_2 x_2.$$

Provided that $x_1 \neq 0$ and $x_2 \neq 0$, this equation can be written as a Hamiltonian equation:

$$\frac{\mathrm{d}}{\mathrm{d}t}\begin{pmatrix} x_1 \\ x_2 \end{pmatrix} = \begin{pmatrix} O & x_1x_2 \\ -x_1x_2 & O \end{pmatrix}\begin{pmatrix} \frac{\partial H}{\partial x_1} \\ \frac{\partial H}{\partial x_2} \end{pmatrix}, \quad H(x_1, x_2) = -a_{21}x_1 - \lambda_2 \ln x_1 + a_{12}x_2 + \lambda_1 \ln x_2,$$

which is different from the standard Hamiltonian equation (1). In fact, this equation cannot be written as (1) globally; the Darboux coordinate system only *locally* exists in general. See Appendix D.

We tested the models other than LNN, which is not applicable because this equation cannot be expressed naturally as a second-order differential equation. We again used simulated solutions for data with 1000 randomly generated initial conditions. Because the Lotka–Volterra model typically describes the population dynamics of certain species, we choose the initial conditions from the uniform distribution on $[0, 1]$. For other details, see Appendix D.

The test time-derivative errors are shown in Table 2. Again, the error by HNN is larger than those by the other models. As seen from the table, the proposed neural symplectic form stably gave better results than the other models.

Regarding the energy errors, the energies are well preserved as shown in Table 3. In particular, the proposed method preserves the energy with the highest accuracy. Besides, predicted orbits are shown in Figure 8. The peaks of the orbits by HNN and the skew matrix learning are smaller than the true trajectory, and the orbit of the NODE is gradually decaying. On the other hand, the proposed neural symplectic form gives the almost identical orbits to the true one.

Thus the hidden symplectic structure of this equation is certainly extracted by the proposed method.

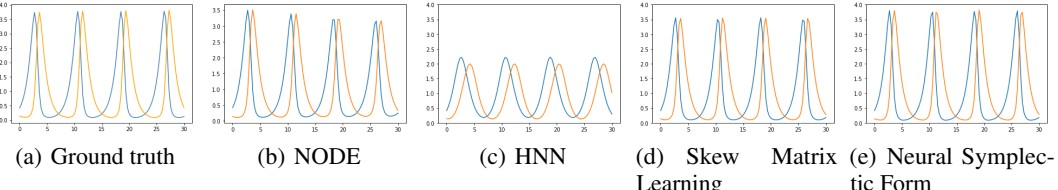

| (a) Ground truth | (b) NODE | (c) HNN | (d) Skew Matrix Learning | (e) Neural Symplectic Form |

Figure 8: Example of the orbits predicted by the proposed and the comparative models for the Lotka–Volterra model. The horizontal axis represents time. $x_1(t), x_2(t)$ are represented: blue ($x_1$) and orange ($x_2$).

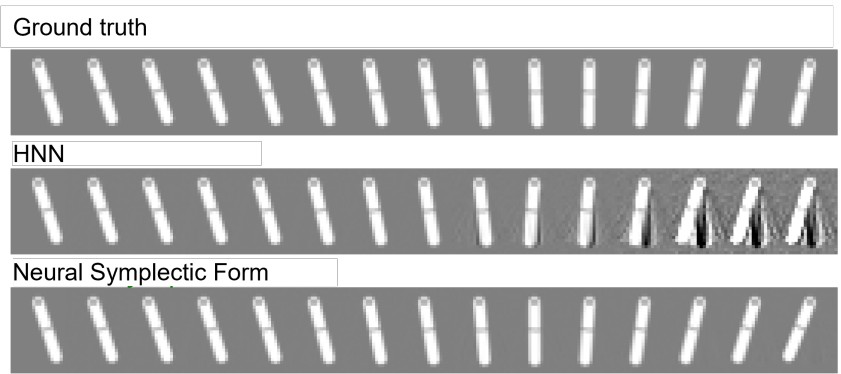

Figure 9: Example of the predicted images by the proposed method and HNN.

**Learning from Images**  As an application of the proposed method, we learned the equation of motion from some images. One way to achieve this is to extract features from the images by using an autoencoder and to learn the equation of motion that the features satisfy [11]. In this case, the extracted features are not supposed to be momenta. The proposed method is suitable for this application because it is independent of the coordinate system.

In this experiment, we first pretrained an autoencoder. Then HNN and the neural symplectic form are applied to learn the dynamics of the features. The final test losses of HNN and the neural symplectic form were 0.161 and 0.060 respectively. The predicted pictures are shown in Figure 9. The last four pictures of HNN are noisy; this implies relatively large errors in the latent space. However, this is only a preliminary test; for example, the performance may depend on the architecture of the autoencoder. Further thorough investigation is needed for this application.

## 5   Concluding Remarks

In this paper, we proposed a method for learning Hamilton equations from data represented on general coordinate systems, which are not restricted to generalized momenta. The key ingredient is the neural symplectic form; we proposed to learn the symplectic 2-form by using neural networks from data, thereby learning this coordinate-free representation. In particular, the Hamilton equation can be represented using a state-dependent skew-symmetric matrix, but not all skew-symmetric matrices are related to the symplectic 2-form. In the proposed method, in order to learn the symplectic 2-form efficiently, the 1-form that derives the symplectic 2-form is learned by the neural networks.

Meanwhile, the proposed method requires the inverse of the skew-symmetric matrix, which may be computationally expensive for modeling large systems. To address this problem, the perturbation theory of the inverse matrix may be applied. For example, it is known that for a matrix $M$ if the norm of $\Delta M$ is small enough, $(M + \Delta M)^{-1} \simeq M^{-1} + M^{-1}\Delta M M^{-1}$ holds. This should be investigated to reduce the computational costs in future work.

## Acknowledgments and Disclosure of Funding

Funding in direct support of this work: JST CREST Grant Number JPMJCR1914, JST PRESTO Grant Number JPMJPR21C7 and JSPS KAKENHI Grant Number 20K11693. The SQUID supercomputer in Osaka University was used for the experiments.

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
