# A Coordinate-free representation of Hamiltonian equations

In terms of geometry, the Hamilton equation is defined as a flow on a symplectic manifold, which is a pair of a manifold and a symplectic 2-form. Because a flow is defined in a coordinate-free form, the Hamilton equation can be defined in a coordinate-free manner as well. In this section, this is explained in more detail. For further information, see, e.g., [3, 18, 20].

Let $\mathcal{M}$ be a manifold, $T\mathcal{M}$ the tangent bundle and $T^*\mathcal{M}$ the cotangent bundle. For each $q \in \mathcal{M}$, $T_q\mathcal{M}$ denotes the tangent space at $q$, which is roughly the space of vectors defined locally at $q$. $T_q^*\mathcal{M}$ is the dual space of $T_q\mathcal{M}$, that is, $T_q^*\mathcal{M}$ is a space of continuous linear maps from $T_q\mathcal{M}$ to $\mathbb{R}$. Differential $k$-forms are the skew-symmetric multilinear maps from $k$ vectors in $T_q\mathcal{M}$ to $\mathbb{R}$. In particular, a 0-form is a function from $\mathcal{M}$ to $\mathbb{R}$ and a 1-form is a vector in the dual space $T_q^*\mathcal{M}$. Suppose that $\mathcal{M}$ is $N$-dimensional and has a local coordinate system $x_1, \ldots, x_N$. Typical 1-forms are $\mathrm{d}x_k$'s, each of which maps a vector $v = (v_1, \ldots, v_N)^\top \in T_q\mathcal{M}$ to $\mathrm{d}x_k(v) = v_k \in \mathbb{R}$. For two 1-forms $\mathrm{d}x_k$ and $\mathrm{d}x_l$, the wedge product $\mathrm{d}x_k \wedge \mathrm{d}x_l$ is a 2-form defined by

$$(\mathrm{d}x_k \wedge \mathrm{d}x_l)(v, w) = v_k w_l - v_l w_k, \quad \text{for all } v = (v_1, \ldots, v_N)^\top, w = (w_1, \ldots, w_N)^\top \in T_q\mathcal{M};$$

in some literature, the wedge product is defined as a constant multiple of the above definition. It follows from the definition that $\mathrm{d}x_k \wedge \mathrm{d}x_l = -\mathrm{d}x_l \wedge \mathrm{d}x_k$, and particularly $\mathrm{d}x_k \wedge \mathrm{d}x_k = 0$.

The exterior derivative $\mathrm{d}$ is a linear operator that computes a certain derivative of differential forms. $\mathrm{d}$ maps a $k$-form to a $(k+1)$-form, and has a characteristic property $\mathrm{d}\mathrm{d} = 0$. For a 0-form $f(q)$, $\mathrm{d}f$ in the coordinate system $x_1, \ldots, x_N$ is defined by

$$\mathrm{d}f = \frac{\partial f}{\partial x_1}\mathrm{d}x_1 + \cdots + \frac{\partial f}{\partial x_N}\mathrm{d}x_N.$$

For a 1-form $\theta = \sum_{k=1}^N f_k \mathrm{d}x_k$, the exterior derivative is

$$\mathrm{d}\theta = \mathrm{d}\sum_{k=1}^N f_k \mathrm{d}x_k = \sum_{l=1}^N \sum_{k=1}^N \frac{\partial f_k}{\partial x_l}\mathrm{d}x_l \wedge \mathrm{d}x_k = \sum_{l<k} \left(\frac{\partial f_k}{\partial x_l} - \frac{\partial f_l}{\partial x_k}\right)\mathrm{d}x_l \wedge \mathrm{d}x_k.$$

For 2 vectors $v = (v_1, \ldots, v_N)^\top, w = (w_1, \ldots, w_N)^\top \in T_q\mathcal{M}$, the value of $\mathrm{d}\theta$ is represented by using a skew matrix $W$

$$\mathrm{d}\theta(v, w) = \sum_{l<k} \left(\frac{\partial f_k}{\partial x_l} - \frac{\partial f_l}{\partial x_k}\right)\mathrm{d}x_l \wedge \mathrm{d}x_k(v, w)$$

$$= \sum_{l<k} \left(\frac{\partial f_k}{\partial x_l} - \frac{\partial f_l}{\partial x_k}\right)(v_l w_k - v_k w_l)$$

$$= (v_1 v_2 \cdots v_N) W \begin{pmatrix} w_1 \\ w_2 \\ \vdots \\ w_N \end{pmatrix}, \quad W = \begin{pmatrix} 0 & \frac{\partial f_2}{\partial x_1} - \frac{\partial f_1}{\partial x_2} & \frac{\partial f_3}{\partial x_1} - \frac{\partial f_1}{\partial x_3} & \cdots \\ \frac{\partial f_1}{\partial x_2} - \frac{\partial f_2}{\partial x_1} & 0 & \frac{\partial f_3}{\partial x_2} - \frac{\partial f_2}{\partial x_3} & \cdots \\ \frac{\partial f_1}{\partial x_3} - \frac{\partial f_3}{\partial x_1} & \frac{\partial f_2}{\partial x_3} - \frac{\partial f_3}{\partial x_2} & 0 & \cdots \\ \vdots & \vdots & \vdots & \ddots \end{pmatrix}.$$

Differential 2-forms are generally represented by using a skew matrix in the same way.

**Definition 1.** *A differential form $\omega$ is closed if $\mathrm{d}\omega = 0$.*

**Definition 2.** *A differential 2-form $\omega$ is non-degenerate if the skew matrix associated with $\omega$ is non-degenerate.*

**Definition 3.** *A symplectic 2-form is a closed and non-degenerate differential 2-form.*

A Hamiltonian equation on a symplectic manifold $\mathcal{M}$ with the symplectic 2-form $\omega$ is defined in a coordinate-free way as

$$\frac{\mathrm{d}u}{\mathrm{d}t} = X_H, \quad \omega(X_H, \cdot) = \mathrm{d}H(\cdot), \tag{8}$$

where $H$ is a Hamiltonian and $X_H$ is a vector field defined by the second equality of the above equations. Note that the differential forms are functions of vectors, which are defined regardless of the coordinate systems. Hence, the model (8) is a coordinate-free representation.

For example, suppose that $\mathcal{M}$ is a 2-dimensional manifold with a local coordinate $(q, p)^\top$. $\mathrm{d}q \wedge \mathrm{d}p$ is a symplectic 2-form on this manifold; in fact, $\mathrm{d}q \wedge \mathrm{d}p$ is obtained as the exterior derivative of 1-form $\theta = q\mathrm{d}p$:

$$\mathrm{d}\theta = \mathrm{d}(q\mathrm{d}p) = \mathrm{d}q \wedge \mathrm{d}p$$

and hence $\mathrm{d}q \wedge \mathrm{d}p$ is closed:

$$\mathrm{d}(\mathrm{d}q \wedge \mathrm{d}p) = \mathrm{d}\mathrm{d}(q\mathrm{d}p) = 0$$

because $\mathrm{d}\mathrm{d} = 0$. The matrix representation of this 2-form is

$$(\mathrm{d}q \wedge \mathrm{d}p)(\begin{pmatrix} q_1 \\ p_1 \end{pmatrix}, \begin{pmatrix} q_2 \\ p_2 \end{pmatrix}) = (q_1 \quad p_1) \begin{pmatrix} 0 & 1 \\ -1 & 0 \end{pmatrix} \begin{pmatrix} q_2 \\ p_2 \end{pmatrix}$$

because

$$(\mathrm{d}q \wedge \mathrm{d}p)(\begin{pmatrix} q_1 \\ p_1 \end{pmatrix}, \begin{pmatrix} q_2 \\ p_2 \end{pmatrix}) = \mathrm{d}q(\begin{pmatrix} q_1 \\ p_1 \end{pmatrix})\mathrm{d}p(\begin{pmatrix} q_2 \\ p_2 \end{pmatrix}) - \mathrm{d}q(\begin{pmatrix} q_2 \\ p_2 \end{pmatrix})\mathrm{d}p(\begin{pmatrix} q_1 \\ p_1 \end{pmatrix}) = q_1 p_2 - q_2 p_1.$$

Meanwhile, $\mathrm{d}H$ is computed as

$$\mathrm{d}H = \frac{\partial H}{\partial q}\mathrm{d}q + \frac{\partial H}{\partial p}\mathrm{d}p.$$

Therefore, by substitution of $\mathrm{d}u/\mathrm{d}t = X_H$ with $u = (q, p)^\top$ into the second equation, the coordinate-free form (8) becomes

$$\mathrm{d}q \wedge \mathrm{d}p \left( \frac{\mathrm{d}}{\mathrm{d}t} \begin{pmatrix} q \\ p \end{pmatrix}, \begin{pmatrix} v_1 \\ v_2 \end{pmatrix} \right) = \mathrm{d}H(\begin{pmatrix} v_1 \\ v_2 \end{pmatrix}) \quad \text{for all } v = \begin{pmatrix} v_1 \\ v_2 \end{pmatrix}$$

of which matrix representation is

$$\left( \frac{\mathrm{d}q}{\mathrm{d}t} \quad \frac{\mathrm{d}p}{\mathrm{d}t} \right) \begin{pmatrix} 0 & 1 \\ -1 & 0 \end{pmatrix} \begin{pmatrix} v_1 \\ v_2 \end{pmatrix} = \frac{\partial H}{\partial q}\mathrm{d}q(\begin{pmatrix} v_1 \\ v_2 \end{pmatrix}) + \frac{\partial H}{\partial p}\mathrm{d}p(\begin{pmatrix} v_1 \\ v_2 \end{pmatrix}) \quad \text{for all } v = \begin{pmatrix} v_1 \\ v_2 \end{pmatrix}.$$

This is equivalent to

$$v_2 \frac{\mathrm{d}q}{\mathrm{d}t} - v_1 \frac{\mathrm{d}p}{\mathrm{d}t} = \frac{\partial H}{\partial q}v_1 + \frac{\partial H}{\partial p}v_2 \quad \text{for all } v_1, v_2,$$

which gives the standard form of the Hamilton equation:

$$\frac{\mathrm{d}q}{\mathrm{d}t} = \frac{\partial H}{\partial p}, \quad \frac{\mathrm{d}p}{\mathrm{d}t} = -\frac{\partial H}{\partial q}.$$

## B De Rham cohomology and the de Rham theorem

From the property $\mathrm{d}\mathrm{d} = 0$ of the exterior derivatives, $\mathrm{Im}\,\mathrm{d} \subset \mathrm{Ker}\,\mathrm{d}$. The difference $\mathrm{Ker}\,\mathrm{d}/\mathrm{Im}\,\mathrm{d}$ is called the de Rham cohomology space.

As is well known, there is a natural duality between differential $k$-forms $\omega$'s and $k$-dimensional integral domains $\Omega$'s in the sense that a real number can be associated with each pairing of $\langle \omega, \Omega \rangle$ in the following way

$$\langle \omega, \Omega \rangle := \int_\Omega \omega.$$

In particular, the Stokes theorem for differential forms

$$\int_\Omega \mathrm{d}\omega = \int_{\partial\Omega} \omega$$

gives the duality between the exterior derivative $\mathrm{d}$ and the boundary operator $\partial$

$$\langle \mathrm{d}\omega, \Omega \rangle = \langle \omega, \partial\Omega \rangle.$$

This duality associates differential forms, the exterior derivative and the cohomology with integral domains, the boundary operator and the homology, respectively. In fact, the boundary operator $\partial$ is defined in such a way that $\partial$ maps $k$-dimensional domain to its $k-1$-dimensional boundary. Because a boundary of a domain is in general a cycle, the boundary of a boundary is 0: $\partial\partial = 0$. This property of the boundary operators is the same as that of the exterior derivative d, and hence the similar space to the cohomology space can be introduced. In fact, because $\mathrm{Im}\,\partial \subset \mathrm{Ker}\,\partial$, we can consider the difference $\mathrm{Ker}\,\partial/\mathrm{Im}\,\partial$. This space is the homology space. The domain contained in the homology space is essentially a "hole," because basically it is a cycle that is not a boundary of any other domain. The de Rham theorem states that there is an isomorphism between the cohomology space and the homology space. In particular, the dimension of the cohomology space is the same as that of the homology space, the number of holes. Thus if the underlying phase space has no hole, the cohomology space vanishes and $\mathrm{Im}\,\mathrm{d} = \mathrm{Ker}\,\mathrm{d}$. When the cohomology space does not vanish, the members of this space must be computed and added to the model. This is possible because this space is finite-dimensional, and hence we can enumerate the members.

## C   Proofs of the theorems

**Proof of Theorem 1**

*Proof.* By the chain rule, we have

$$\frac{\mathrm{d}H_{\mathrm{NN}}(u)}{\mathrm{d}t} = \nabla H_{\mathrm{NN}} \cdot \frac{\mathrm{d}u}{\mathrm{d}t}.$$

Substituting (3) into the above equation, we get

$$\frac{\mathrm{d}H_{\mathrm{NN}}(u)}{\mathrm{d}t} = \nabla H_{\mathrm{NN}} \cdot W_{u,\mathrm{NN}}^{-\top}\nabla H_{\mathrm{NN}} = 0$$

because $W_{u,\mathrm{NN}}^{-\top}$ is skew-symmetric and for any skew-symmetric matrix $M$ and for any vector $v$

$$v \cdot Mv = 0,$$

which is confirmed by

$$v \cdot Mv = v^\top Mv = (v^\top Mv)^\top = v^\top M^\top v = -v^\top Mv = -v \cdot Mv$$

because $M$ is skew-symmetric and $v^\top Mv$ is a $1 \times 1$ matrix, for which $(v^\top Mv)^\top = v^\top Mv$ holds. The above equality gives $2v \cdot Mv = 0$.  □

**Proof of Theorem 3**   First, we show the precise statement of Theorem 3, in which we denote by $\Sigma(\sigma)$ the space of the neural networks with the activation function $\sigma$:

$$\Sigma(\sigma) = \{g : \mathbb{R}^r \to \mathbb{R} \mid g(x) = \sum_i^q \beta_i\sigma(\gamma_i^\top x + \alpha_i), \alpha_i \in \mathbb{R}, \beta_i \in \mathbb{R}, \gamma_i \in \mathbb{R}^r\}.$$

**Theorem .** *Suppose that the Hamilton equation to be learned can be represented in the form* (2) *using a Hamiltonian $H \in W^{1,p}$ and a symplectic 2-form $\omega$ that is derived from a 1-form $\theta \in W^{\mathrm{d},p}$ in the sense that*

$$\mathrm{d}\theta = \omega.$$

*Suppose also that the phase space is compact. The model* (4) *with neural networks in $\Sigma(\sigma)$, of which activation function $\sigma$ is in $C^\infty$ and does not vanish everywhere, has the universal approximation property in the sense that $\theta$ and $H$ can be approximated by the neural networks with arbitrary accuracy.*

*Proof.* In the spaces $W^{1,p}$ and $W^{\mathrm{d},p}$, the spaces of $C^\infty$ functions $C^\infty \cap W^{1,p}$ and $C^\infty \cap W^{\mathrm{d},p}$ are respectively dense [14]. Therefore, if neural networks used in the model admit the universal approximation property to $C^\infty$ functions, then the universal approximation theorem for the differential forms holds. Meanwhile, regarding the approximation of $C^\infty$ functions, the following theorem is known.

**Theorem** (Hornik et al., 1990)**.** *If the activation function $\sigma \neq 0$ belongs to $S_p^m(\mathbb{R})$ for an integer $m \geq 0$, then $\Sigma(\sigma)$ is $m$-uniformly dense in $C^\infty(K)$, where $K$ is any compact subset of $\mathbb{R}^N$.*

$S_p^m(\mathbb{R})$ is the Sobolev space, which is roughly functions with up to $m$th (weak) derivatives with the bounded $L^p$ norm. $W^{1,p}$ and $W^{d,p}$ are Sobolev spaces of differential forms; see [2, 14] for the definitions and the properties. Hence, if the activation function $\sigma$ of the hidden layer is in $C^\infty \subset S_p^m(\mathbb{R})$ and does not vanish everywhere, then for any $C^\infty$ function, there exists a neural network that approximates this function. Since it is assumed that $\sigma$ is $C^\infty$ and does not vanish everywhere, we need to prove that $\sigma$ and its derivatives are in $L^p$. Because the phase space is assumed to be compact and the activation functions are smooth, the outputs of the neural network and hidden layers are also compact. Hence, the activation function $\sigma$ is essentially used on the compact domains. Therefore, $\sigma$ can be restricted to such domains so that $\sigma$ is in $L^p$. This completes the proof. □

## D Supplemental note on numerical experiments

In this appendix, we make some comments on the datasets and the results of the experiments with some additional experiments. In particular, to examine the results in more detail, we plotted the errors for each experiment as a function of time for the well-performed models, i.e., NODE, the skew matrix learning and the neural symplectic form.

In the experiments, we implemented all code using Python v3.8.5 with libraries; numpy v1.21.2, scipy v1.7.1, and PyTorch v1.9.1. We performed all experiments on NVIDIA A100. We employed a neural network with two hidden layers of 200 units and the tanh activation function for modeling the energy function, the skew-symmetric matrix, and the 1-form for the neural symplectic form. We used 80 percent of the collected data for training and the remaining for test. The collected data were normalized so that most of them were in the range $[-1, 1]$; however, the errors in the main text are given in the original physical scale. We trained each model 10 times using the Adam optimizer with a learning rate of $10^{-3}$ for 2000 iterations. Only for LNN, we set the learning rate to $10^{-4}$ due to an instability of learning. We explain this in detail below.

**On the performances of LNNs** In our experiments, although LNNs should be capable of modeling the targets, LNNs sometimes did not work well. Figure 10 shows an example of the histories of the training loss of an LNN for the double pendulum data when the learning rate is set to $10^{-3}$. As shown in the figure, the behavior of the loss function was not stable.

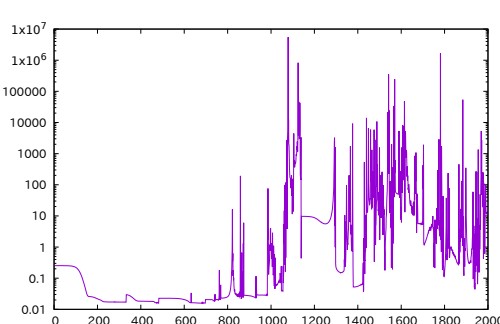

This may be due to the lack of enough data; in fact we use a much smaller data than Cranmer et al. [7]. In addition, the neural network in Cranmer et al. [7] had three hidden layers with 500 units, whereas our neural network had two hidden layer with 200 units. The variance of the randomly initialized weight parameters was

Figure 10: An example of the histories of the training loss of LNN for the double pendulum data when the learning rate is set to $10^{-3}$.

adjusted for each layer using more than 200 preliminary trainings. The dataset was composed of 600,000 orbits, whereas we used only 2,000 orbits. These differences should make the learning of LNN more stable than our experiments.

Another possibility is the non-uniqueness of the Lagrangian. In LNNs, the Lagrangian of the target dynamics is learned from the given data. Suppose that a Lagrangian $\mathcal{L}_{NN}$, which is learned from the data, fits the given data. Then, the dynamics must be described by the Euler–Lagrange equation

$$\frac{\partial \mathcal{L}_{NN}}{\partial q} - \frac{d}{dt}\frac{\partial \mathcal{L}_{NN}}{\partial \dot{q}} = 0.$$

However, there are many Lagrangians that give the same Euler–Lagrange equation. In fact, the Euler–Lagrange equation is derived by the variational principle, in which the stationary value of the

action integral

$$S = \int_0^T \mathcal{L}(q, \dot{q}) \mathrm{d}t$$

with both ends fixed; however, even if the learned Lagrangian gives a stationary point, other Lagrangians can also give a stationary point. For example, for any smooth function $f(q)$, another Lagrangian $\tilde{\mathcal{L}} := \mathcal{L} + \mathrm{d}f/\mathrm{d}t$ gives the same Euler–Lagrange equation because the action integral associated with $\tilde{\mathcal{L}}$

$$\tilde{S} = \int_0^T \left( \mathcal{L}(q, \dot{q}) + \frac{\mathrm{d}f}{\mathrm{d}t} \right) \mathrm{d}t = \int_0^T \mathcal{L}(q, \dot{q}) \mathrm{d}t + f(q(T)) - f(q(0)) = S + f(q(T)) - f(q(0))$$

takes the same stationary value as the original action integral $S$ when both ends fixed. For example, two Lagrangians $\mathcal{L}$ and $\tilde{\mathcal{L}} = \mathcal{L} + q\dot{q}$ gives the same Euler–Lagrange equation because $q\dot{q} = \frac{1}{2}\frac{\mathrm{d}(q^2)}{\mathrm{d}t}$; the Euler–Lagrange equation of $\tilde{\mathcal{L}}$ is

$$\frac{\partial \tilde{\mathcal{L}}}{\partial q} - \frac{\mathrm{d}}{\mathrm{d}t}\left( \frac{\partial \tilde{\mathcal{L}}}{\partial \dot{q}} \right) = \frac{\partial \mathcal{L}}{\partial q} + \dot{q} - \frac{\mathrm{d}}{\mathrm{d}t}\left( \frac{\partial \mathcal{L}}{\partial \dot{q}} + q \right) = \frac{\partial \mathcal{L}}{\partial q} - \frac{\mathrm{d}}{\mathrm{d}t}\frac{\partial \mathcal{L}}{\partial \dot{q}},$$

which is the Euler–Lagrange equation of $\mathcal{L}$. In LNNs, the Lagrangian is directly modeled by neural networks; however, because the Lagrangian is not uniquely determined, even after a good Lagrangian is found, the learning algorithm goes on to find another Lagrangian which is possibly better. This is considered the reason for the instability of the learning processes of LNNs.

As a matter of fact, the loss function of LNN was unstable, but can be small. Therefore, in the numerical experiments, the model that achieved the minimum value of the histories was used to analyze the behavior of the models. In addition, the learning rate was set to a smaller value so that the training process could be more stable.

**Mass-Spring System** Firstly, we explain the mass-spring system depicted in Figure 3, which is used in the first experiment. As the data, we used numerical solutions to

$$\frac{\mathrm{d}}{\mathrm{d}t}\begin{pmatrix} q_1 \\ q_2 \\ v_1 \\ v_2 \end{pmatrix} = \begin{pmatrix} v_1 \\ v_2, \\ -\frac{k_1}{m_1}(q_1 - l_1) + \frac{k_2}{m_1}(q_2 - q_1 - l_2) \\ -\frac{k_2}{m_2}(q_2 - q_1 - l_2) \end{pmatrix}, \tag{9}$$

with the parameters $k_1 = 3.0$, $k_2 = 5.0$, $l_1 = 1.0$, $l_2 = 1.0$, $m_1 = 1.0$, $m_2 = 2.0$. The initial values are randomly sampled from the standard normal distribution. The numerical solutions are computed on the time interval $[0, 5]$. For each numerical orbit, 100 solutions are sampled at uniform time intervals. The numerical solutions are by using SciPy odeint with the default setting.

Note that it should be impossible to write this equation as the standard form of Hamiltonian equation (1) with a certain energy function; for example, when the system has just one mass point and the equation of motion is given by

$$\frac{\mathrm{d}}{\mathrm{d}t}\begin{pmatrix} q_1 \\ v_1 \end{pmatrix} = \begin{pmatrix} v_1 \\ -\frac{k_1}{m_1}(q_1 - l_2) \end{pmatrix},$$

this can be transformed into a Hamiltonian system

$$\frac{\mathrm{d}}{\mathrm{d}t}\begin{pmatrix} q_1 \\ v_1 \end{pmatrix} = \begin{pmatrix} 0 & 1 \\ -1 & 0 \end{pmatrix} \nabla \tilde{H},$$

$$\tilde{H} = \frac{v_1^2}{2m_1'} + k_1'(q_1 - l_1)^2$$

with $k_1' = k_1/m_1, m_1'$ and the mass $m_1'$ is 1. Hence, for this system, the Hamiltonian neural networks are applicable without knowledge of $m_1$. However, for the above system (9), such a transformation cannot be applied.

Figure 11 shows the time evolution of the errors for NODE, the skew matrix learning and the neural symplectic form. The corresponding orbits are shown in Figure 12. In this case, the prediction of

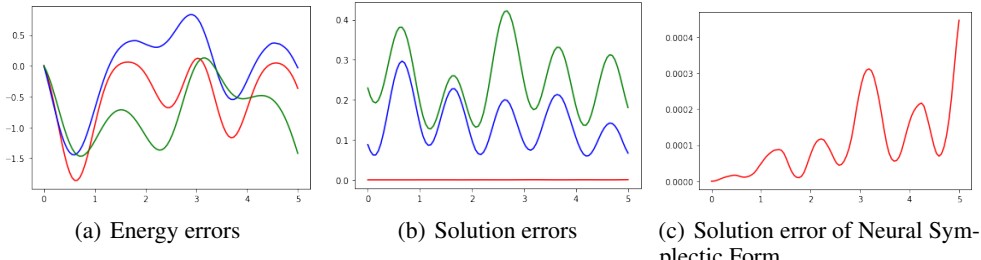

(a) Energy errors  (b) Solution errors  (c) Solution error of Neural Symplectic Form

Figure 11: Time evolution of the energy and solution errors obtained by NODE, the skew matrix learning and the neural symplectic form for the mass-spring test. The horizontal axis represents time. The energy error shows the difference from the true energy, and the solution error shows the MSEs. Since the solution error by the neural symplectic form was tiny, the enlarged graph is also shown.

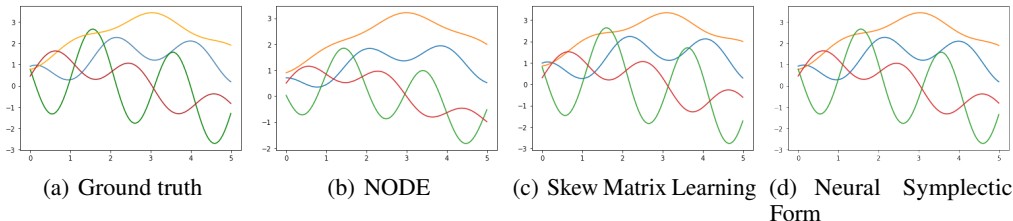

(a) Ground truth  (b) NODE  (c) Skew Matrix Learning  (d) Neural Symplectic Form

Figure 12: Example of the orbits predicted by the trained models for the mass-spring test. The horizontal axis represents time. Each component of $u(t) = (q_1(t), v_1(t), q_2(t), v_2(t))$ is represented: blue ($q_1$), green ($v_1$), orange ($q_2$), and red ($v_2$).

the neural symplectic form was very accurate. This may be related to the fact that the proposed method is able to preserve the conservation laws other than energy. In fact, in the research field of physical simulation, it is known that for such a small system, the computational accuracy can be often improved if additional conservation laws are preserved. As for the energy error, there is not much difference between the methods. The energy error appears to be large compared to the solution error; this should be due to the numerical integration errors in the computation of the predicted orbits.

**Double Pendulum**  Secondly, we explain the double pendulum in the second experiment:

$$\frac{\mathrm{d}\theta_1}{\mathrm{d}t} = \phi_1, \quad \frac{\mathrm{d}\theta_2}{\mathrm{d}t} = \phi_2,$$

$$\frac{\mathrm{d}\phi_1}{\mathrm{d}t} = \frac{g(\sin\theta_2 \sin(\theta_1 - \theta_2) - \frac{m_1+m_2}{m_2}\sin(\theta_1)) - (l_1\theta_1^2 \cos(\theta_1 - \theta_2) + l_2\theta_2^2)\sin(\theta_1 - \theta_2)}{l_1(\frac{m_1+m_2}{m_2} - \cos^2(\theta_1 - \theta_2))},$$

$$\frac{\mathrm{d}\phi_2}{\mathrm{d}t} = \frac{\frac{g(m_1+m_2)}{m_2}(\sin\theta_1 \cos(\theta_1 - \theta_2) - \sin(\theta_2)) - (\frac{l_1(m_1+m_2)}{m_2}\theta_1^2 + l_2\theta_2^2 \cos(\theta_1 - \theta_2))\sin(\theta_1 - \theta_2)}{l_2(\frac{m_1+m_2}{m_2} - \cos^2(\theta_1 - \theta_2))}.$$

The energy function of this system is

$$H = \frac{1}{2}(m_1 + m_2)l_1^2\phi_1^2 + \frac{1}{2}m_2l_2^2\phi_2^2 + m_2l_1l_2\phi_1\phi_2 \cos(\theta_1 - \theta_2) + gm_2l_2 \cos(\theta_2)$$
$$+ g(m_1 + m_2)l_1 \cos\theta_1$$

and the Lagrangian is

$$\mathcal{L} = \frac{1}{2}(m_1 + m_2)l_1^2\phi_1^2 + \frac{1}{2}m_2l_2^2\phi_2^2 + m_2l_1l_2\phi_1\phi_2 \cos(\theta_1 - \theta_2) - gm_2l_2 \cos(\theta_2)$$
$$- g(m_1 + m_2)l_1 \cos\theta_1,$$

which derives the generalized momentum

$$p_1 = \frac{\partial \mathcal{L}}{\partial \phi_1} = (m_1 + m_2)l_1^2\phi_1 + m_2l_1l_2\phi_2 \cos(\theta_1 - \theta_2),$$

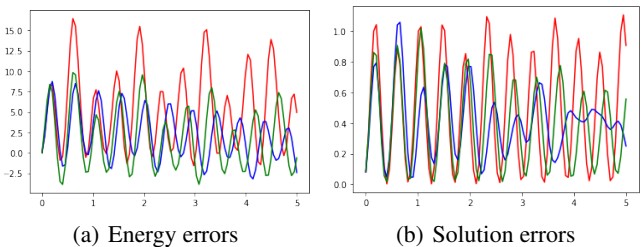

|     |     |
| --- | --- |
| (a) Energy errors | (b) Solution errors |

Figure 13: Time evolution of the energy and solution errors obtained by NODE, the skew matrix learning and the neural symplectic form for the double-pendulum test. The errors are represented: blue (skew matrix learning), green (NODE), red (neural symplectic form). The horizontal axis represents time. The energy error shows the difference from the true energy, and the solution error shows the MSEs.

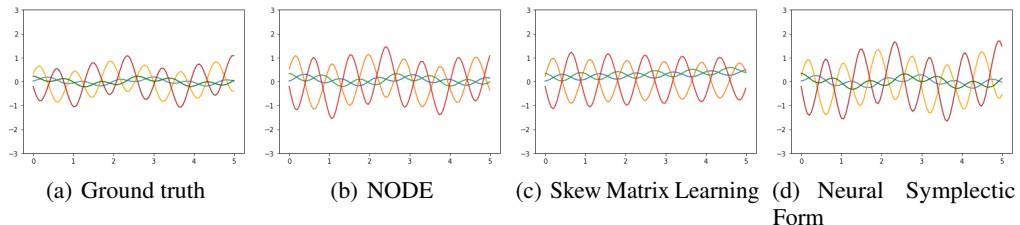

|     |     |     |     |
| --- | --- | --- | --- |
| (a) Ground truth | (b) NODE | (c) Skew Matrix Learning | (d) Neural Symplectic Form |

Figure 14: Example of the orbits predicted by the trained models for the double-pendulum test. The horizontal axis represents time. Each component of $u(t) = (\theta_1(t), \phi_1(t), \theta_2(t), \phi_2(t))$ is represented: blue ($\theta_1$), orange ($\phi_1$), green ($\theta_2$), and red ($\phi_2$).

$$p_2 = \frac{\partial \mathcal{L}}{\partial \phi_2} = m_2 l_2^2 \phi_2 + m_2 l_1 l_2 \phi_1 \cos(\theta_1 - \theta_2).$$

Because the generalized momenta are written in the very complicated forms, it is difficult to prepare data in these forms.

We collected data by solving (6) with the parameters $l_1 = l_2 = 1.0$, $m_1 = 1$, $m_2 = 2$, and $g = 9.8$ using 2000 initial conditions randomly generated from the standard normal distribution. The batch-size was set to 1000.

Firstly, we show the time evolution of the errors for NODE, the skew matrix learning and the neural symplectic form in Figure 13 along with the corresponding orbits in Figure 14. The errors are not significantly different; however, the errors for NODE and the skew matrix learning are relatively small. Although these methods are not suitable for long-term predictions, they are effective for short-term predictions.

Secondly, we also performed a similar test to the main paper for a more chaotic orbit using NODE and the neural symplectic form. Examples of the predicted orbits are shown in Figure 15. Note that in this experiment, we used models trained with the same data as the experiment in the main paper, and this data may not contain many chaotic trajectories.

Because the results of the NODE often diverged as shown in this figure we omit the quantitative results for this test. Although the predicted orbits are not so similar to the true one, the proposed method typically kept oscillating. The worse performance may be due to the lack of enough data on the behaviors when the angle became large. In fact, a chaotic double pendulum can rotate many times, and its angle can be very large; however, our data are generated by simulating the orbits from relatively small initial conditions. For better performance, the periodic structure of the phase space as a Lie group should be integrated into the model for efficient training.

**Lotka–Volterra equation** Thirdly, we investigated the models for the Lotka–Volterra equation,

$$\frac{\mathrm{d}x_1}{\mathrm{d}t} = a_{12}x_1x_2 + \lambda_1 x_1, \quad \frac{\mathrm{d}x_2}{\mathrm{d}t} = a_{21}x_1x_2 + \lambda_2 x_2.$$

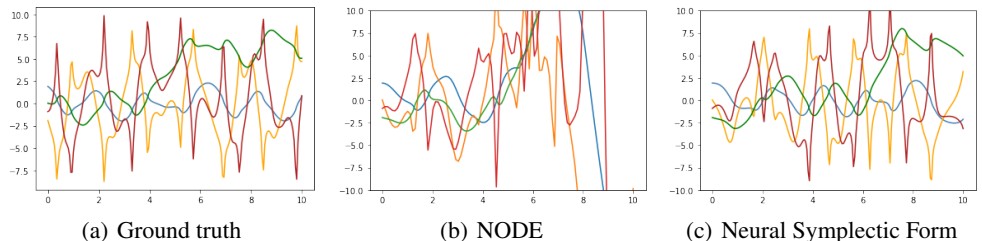

| (a) Ground truth | (b) NODE | (c) Neural Symplectic Form |

Figure 15: Example of the orbits predicted by the trained models for the double pendulum with a chaotic behavior. The horizontal axis represents time. Each component of $u(t) = (q_1(t), v_1(t), q_2(t), v_2(t))$ is represented: blue ($q_1$), green ($v_1$), orange ($q_2$), and red ($v_2$).

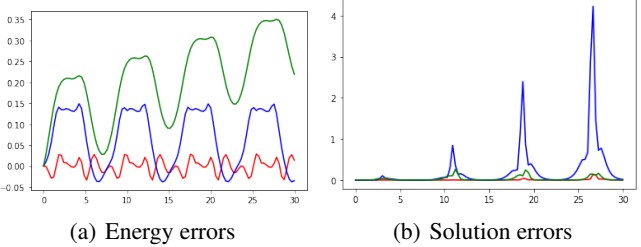

| (a) Energy errors | (b) Solution errors |

Figure 16: Time evolution of the energy and solution errors obtained by NODE, the skew matrix learning and the neural symplectic form for the Lotka–Volterra test. The errors are represented: blue (skew matrix learning), green (NODE), red (neural symplectic form). The horizontal axis represents time. The energy error shows the difference from the true energy, and the solution error shows the MSEs.

which is a Hamiltonian equation but does not admit the standard Hamiltonian form (1) in general. For example, if the first equation can be written as

$$\frac{\mathrm{d}x_1}{\mathrm{d}t} = \frac{\partial H}{\partial x_2}$$

with a certain function $H(x_1, x_2)$, $H(x_1, x_2)$ should be of the form

$$H(x_1, x_2) = \frac{a_{12}}{2} x_1 x_2^2 + \lambda_1 x_1 x_2 + f(x_1)$$

with a function $f$. However, in that case it must hold that

$$\frac{\partial H(x_1, x_2)}{\partial x_1} = \frac{a_{12}}{2} x_2^2 + \lambda_1 x_2 + \frac{\mathrm{d}f(x_1)}{\mathrm{d}x_1},$$

which cannot be in general the right-hand side of the second equation of the Lotka–Volterra equation for any $f$. In the experiment, we checked if this unknown symplectic structure can be extracted from the data or not.

In the experiment, because the state variables are not the pair of $q$ and $\dot{q}$, LNN is not applicable to this equation. Hence, we tested NODE, HNN, the skew matrix learning and the neural symplectic form. We set the parameters as $a_{12} = -1, a_{21} = -1, \lambda_1 = 1$ and $\lambda_2 = 1$. As the data, 1000 orbits on the time interval $[0, 5]$ are numerical computed by using SciPy odeint. Initial conditions are sampled from the uniform distribution on $[0, 1]$. Each orbit contains 100 numerical solutions at a uniform sampling rate. The data are normalized so that they are in $[0, 1]$.

We show the time evolution of the errors for NODE, the skew matrix learning and the neural symplectic form in Figure 16. The corresponding orbits are shown in Figure 17. Since the peaks of the orbit of the NODE is gradually decaying, the energy error of the NODE is increasing. Regarding the solution errors, although the predicted orbit of the skew matrix learning is not so different from the ground truth, the error becomes very large. This is due to the error in the velocity of the oscillation. This error causes the position of the peaks to deviate from its true position, resulting in a large error.

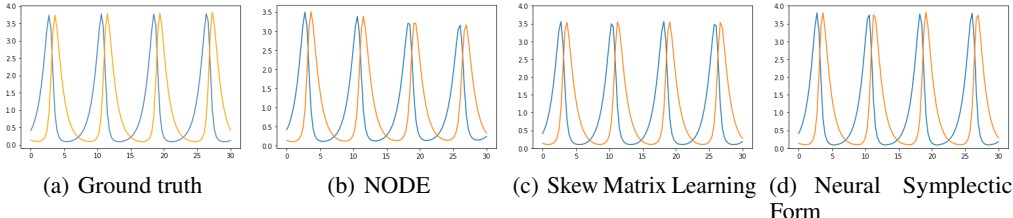

(a) Ground truth  (b) NODE  (c) Skew Matrix Learning  (d) Neural Symplectic Form

Figure 17: Example of the orbits predicted by the trained models for the Lotka–Volterra test. The horizontal axis represents time. $x_1(t), x_2(t)$ are represented: blue ($x_1$) and orange ($x_2$).

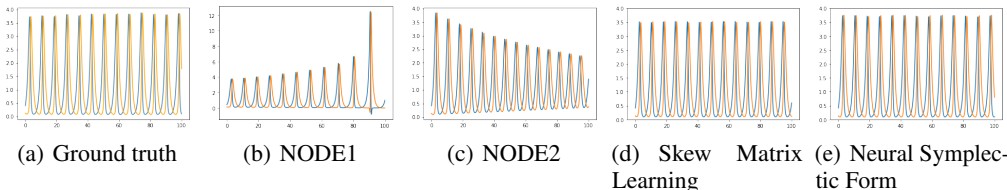

(a) Ground truth  (b) NODE1  (c) NODE2  (d) Skew Matrix Learning  (e) Neural Symplectic Form

Figure 18: Example of the orbits during a long period ($t = 100$) predicted by the trained models for the Lotka–Volterra test. NODE1 and NODE2 are results of two NODE models trained with a different random seed. The horizontal axis represents time. $x_1(t), x_2(t)$ are represented: blue ($x_1$), orange ($x_2$).

Table 4: Energy errors.

|  | NODE | Skew Matrix Learning | Neural Symplectic Form (proposed) |
|---|---|---|---|
| Lotka–Volterra ($t = 100$) | $42.951 \pm 56.005$ | $\underline{0.0735 \pm 0.0764}$ | $\mathbf{0.0704 \pm 0.0597}$ |

The best and second best results of the true energies are emphasized by bold and underlined fonts, respectively. The differences of the true energy functions at $t = 100$ and $t = 0$ are shown. 10 orbits were simulated using randomly generated initial values.

In addition to the experiments shown in the main paper, we also performed another test in which long term behaviors are predicted. In this case, we compared the well-performed models: NODE, skew matrix learning and the neural symplectic form. The predicted orbits are shown in Figure 18. The two NODE results are obtained by the two NODE models trained with a different random seed. Because NODE does not satisfy the energy conservation law, the predicted orbits by these models gradually decay or diverge as expected. Meanwhile, the orbits by the other two models keep oscillating. The energy errors are shown in Table 4. While the error of the NODE is very large, the results by the other two are almost the same. The error of the skew matrix learning is slightly larger; this is probably due to the lower height of the peaks.

**Learning from images**  In this experiment, we used the HNN code and data (`https://github.com/greydanus/hamiltonian-nn`, Apache 2.0 License) almost as-is; we used a pre-trained autoencoder with two hidden layers of 200 units and the ReLU activation function. We trained the autoencoder using the Adam optimizer with a learning rate of $10^{-4}$ for 150000 iterations. The test error of the autoencoder was 3.81e-03. The neural networks in the HNN and the neural symplectic forms have two hidden layers of 200 units and the tanh activation function. These networks are trained for 100000 steps; in our experiment it took a long time for the neural symplectic form to find an appropriate symplectic form. The original HNN code includes a loss function that measures how close the latent space is to the canonical coordinates in order to make it closer to the canonical coordinates. However, in our experiment, we trained without this loss function. The other settings are the same as in the similar experiment of HNN. See [11] for details.