# OpenReview forum: "Neural Symplectic Form: Learning Hamiltonian Equations on General Coordinate Systems"
_NeurIPS.cc/2021/Conference — NeurIPS 2021 Spotlight_

### Official Review · Reviewer_XXhP · 2021-07-12

**Rating:** 7
**Confidence:** 4

**Summary:**

This paper considers the learning of non-canonical Hamiltonian dynamics from time-series data. Canonical Hamiltonian systems in Euclidean spaces correspond to $\dot{x}=J \nabla H(x)$ where $J=[0,  I; -I, 0]$ is a constant matrix corresponding to the canonical symplectic form. Non-canonical Hamiltonian systems are in the same form, however $J$ can be $x$ dependent, and it has to be a skew-symmetric matrix satisfying Leibniz’s rule and Jacobi identity. This paper studies the Euclidean case, which has nice enough topology so that $J:\mathbb{R}^{d}\to \mathbb{R}^{d\times d}$ satisfying these conditions can be expressed using a vector field $Y: \mathbb{R}^{d}\to \mathbb{R}^d$ instead, which is no longer constrained. The proposed method represents $H$ and $Y$ by neural networks and trains them to match the data. Some numerical investigations are provided.

**Main Review:**

The method is new and interesting. Although it needs a nice cohomology like that of the Euclidean space, the setup corresponds to sufficiently many important problems. The paper is well written and the informal descriptions are very helpful to general readers.

While I like the continuous (in time) part of the paper, I hope the authors could agree that discretization matters very much when the method is actually implemented. This is something that the current version can improve upon. For example, the formulation of the method is summarized inbetween Line 189 and 190, but how is $du/dt$ estimated? After the continuous dynamics is learned, how is prediction done? In fact, these questions are central to many of the recent advances in this area, some of them already cited, such as [20, 21], but two important references are missed. Namely, [Jin et al. SympNets: Intrinsic structure-preserving symplectic networks for identifying Hamiltonian systems. Neural Networks 2020] and [Chen & Tao. Data-driven Prediction of General Hamiltonian Dynamics via Learning Exactly-Symplectic Maps. ICML 2021]. These aspects should be discussed for the sake of the readers.

In addition, [12] was cited but not sufficiently discussed. However, it is closely related work and their similarity and difference should be clarified.

Continuing along the line of discretization, it was written that “A numerical integration of the Hamiltonian system is known to destroy the symplectic structure and does not conserve the Hamiltonian H.” This is incorrect. Symplectic integrators exist; see for instance [Hairer Lubich Wanner. Geometric Numerical Integration. 2006] for separable Hamiltonians (and much more) and [Tao. Explicit symplectic approximation of nonseparable Hamiltonians. 2016] for the rest of the picture. Regarding conserving the Hamiltonian, though, one cannot have simultaneous exact conservation of symplecticity and energy in constant step integrations, as the Ge-Marsden theorem gives.

In addition, I would like to understand two things:

Firstly, the proposed method is better than skew matrix learning and much better than HNN, according to the mass spring experiment and the double pendulum experiment, but why? For these two problems, the coordinates are already canonical, so I was expecting the new method to degenerate to HNN and give comparable performance, but it is not the case. What are the learned symplectic 2-forms in these two cases? If it is because of different network architecture and/or training details, the comparison needs to be made fair.

Secondly, how is the invertibility of $\tilde{W}$ ensured?


**Time Spent Reviewing:**

3

---

> ### Author Response · Authors · 2021-08-10
> **Response to Reviewer**
>
>
> **Q1a: While I like the continuous (in time) part of the paper, I hope the authors could agree that discretization matters very much when the method is actually implemented. This is something that the current version can improve upon. For example, the formulation of the method is summarized inbetween Line 189 and 190, but how is du/dt estimated? After the continuous dynamics is learned, how is prediction done?**
>
> Thank you very much for your kind review.
>
> We agree with you in that discretization certainly matters, and combination of the proposed model and the structure-preserving methods, such as symplectic integrators, variational integrators and also energy-preserving integrators, surely improves the results.
>
> However, in order to focus on learning the symplectic form, which is the main proposal, we prefer not to bias the focus of the discussion towards discretization methods since there are too many integrators to be tested as listed above. Hence, we consider simple situations where the data of du/dt are given by the continuous true models and the classical Runge-Kutta method (ode45) is used for prediction.
>
> **Q1b: In fact, these questions are central to many of the recent advances in this area, some of them already cited, such as [20, 21], but two important references are missed. Namely, [Jin et al. SympNets: Intrinsic structure-preserving symplectic networks for identifying Hamiltonian systems. Neural Networks 2020] and [Chen & Tao. Data-driven Prediction of General Hamiltonian Dynamics via Learning Exactly-Symplectic Maps. ICML 2021]. These aspects should be discussed for the sake of the readers.**
>
> In [Jin et al., Neural Networks, 2020] and [Chen & Tao, ICML, 2021], the symplectic maps are learned from data under the assumption of the standard symplectic matrix $J = \\pmatrix{O & I \\\\ -I & O}$. Because the concept of neural symplectic form is independent from the discretization methods and learning objectives, it is available with these models after a slight modification. As you suggested, discussion on the relationship to these models is important; we will discuss them by citing these papers in the final version.
>
> **Q1c: Continuing along the line of discretization, it was written that “A numerical integration of the Hamiltonian system is known to destroy the symplectic structure and does not conserve the Hamiltonian H.” This is incorrect. Symplectic integrators exist; see for instance [Hairer Lubich Wanner. Geometric Numerical Integration. 2006] for separable Hamiltonians (and much more) and [Tao. Explicit symplectic approximation of nonseparable Hamiltonians. 2016] for the rest of the picture. Regarding conserving the Hamiltonian, though, one cannot have simultaneous exact conservation of symplecticity and energy in constant step integrations, as the Ge-Marsden theorem gives.**
>
> Sorry for the confusion. We used "numerical integrator" to mean "non-geometric numerical integrator". As you pointed out, geometric numerical integrators including symplectic integrators either preserve the symplectic structure or conserve the energy. We will clarify the explanations by citing the suggested references in the final version.
>
> **Q2: In addition, [12] was cited but not sufficiently discussed. However, it is closely related work and their similarity and difference should be clarified.**
>
> In [12], discrete-time models for Poisson systems is proposed. Poisson systems are extension of Hamiltonian systems, where the skew symmetric matrix can be degenerate. In particular, in [12], dynamics with state-dependent skew symmetric matrices are leaned. The main difference between the method in [12] and the proposed one is that the method in [12] uses coordinate transformations for learning the dynamics in the latent space, while our method does not. Our method has two advantages: (1) interpretability: the equation of motion is defined not in the latent space but in the observable data space, and (2) uniqueness of the model: the latent spaces associated with the given data may not be unique. In fact, a standard Hamiltonian equation
> $$
> \frac{\mathrm{d}u_1}{\mathrm{d}t} = \pmatrix{O & I \\\\ -I & O}\frac{\partial H_1}{\partial u_1}
> $$
> may be able to be transformed into another Hamiltonian system of the same form
> $$
> \frac{\mathrm{d}u_2}{\mathrm{d}t} = \pmatrix{O & I \\\\ -I & O}\frac{\partial H_2}{\partial u_2}
> $$
> by a canonical transformation. This may make learning the model inefficient. We will add this discussion in the final version.
>
> **Q3: Firstly, the proposed method is better than skew matrix learning and much better than HNN, according to the mass spring experiment and the double pendulum experiment, but why? For these two problems, the coordinates are already canonical, so I was expecting the new method to degenerate to HNN and give comparable performance, but it is not the case. What are the learned symplectic 2-forms in these two cases? If it is because of different network architecture and/or training details, the comparison needs to be made fair.**
>
> In the mass spring test and the double pendulum test, the coordinates $(q, \dot{q})$ are not canonical, while HNN assumes the canonical coordinates. More precisely, these systems cannot be written as
> $$
> \frac{\mathrm{d}}{\mathrm{d}t}\pmatrix{q \\\\ \dot{q}} = \\pmatrix{O & I \\\\ -I & O}\\pmatrix{\frac{\partial H}{\partial q} \\\\ \frac{\partial H}{\partial \dot{q}} }
> $$
> with an energy function $H(q, \dot{q})$. This is the reason why HNN failed; HNN does not admit the approximation property in general if the data are given in the coordinate $(q, \dot{q})$. In particular, it is not because of different network architecture nor training details.
>
> For the Hamilton equation to be written as the above canonical form, the coordinate $(q, p)$​ must be used. Please note that $p$​ is the generalized momentum, which is not $\dot{q}$​ multiplied by the mass. For example, $p$​ for the double pendulum is shown in L247. For detailed explanation for the mass-spring system, please see L.578--L.584 in Supplemental Materials.
>
> HNN cannot learn these dynamics without knowledge of the analytical expression of the generalized momentum; however, this cannot be expected to known in advance in practice because these are defined by using the energy function, which is exactly what we want to learn from data. Removing this limitation of HNN is the main objective of this paper.
>
> As suggested by other reviewers, for better interpretability of the results, we will replace the results in the submitted paper with the ones below. We have performed additional experiments in the standard setting of HNNs, in which the networks have two hidden-layers with 200 units and the number of training epochs is 2000. In addition, to reduce the effects of rounding error, we performed the experiments with double precision using Titan V GPUs. The results are summarized as follows, in which the approximation ability of HNN are shown to be limited again.
>
> |                     |   skew matrix learning    |    neural symplectic form     |            NODE             |            HNN            |         LNN         |          LNN(best)          |
> |:--------------------|:-------------------------:|:-----------------------------:|:---------------------------:|:-------------------------:|:-------------------:|:---------------------------:|
> | **mass spring**     | 9.614e-05 $\pm$ 8.525e-05 | **1.214e-07** $\pm$ 5.208e-08 |  3.912e-07 $\pm$ 3.384e-07  | 7.923e-05 $\pm$ 6.286e-07 | 7.786 $\pm$ 23.357  | *1.381e-07* $\pm$ 1.050E-07 |
> | **double pendulum** | 3.775e-04 $\pm$ 2.964e-04 | **3.909e-05** $\pm$ 9.704e-06 | *1.882e-04* $\pm$ 2.555e-05 | 3.821e-03 $\pm$ 5.100e-04 | 23.210 $\pm$ 36.156 |   2.092 e-2 $\pm$ 1.116e-4   |
>
> **Q4: Secondly, how is the invertibility of**  **$\tilde{W}$** **ensured?**
>
> The learned 1-form and hence the 2-form and $\tilde{W}$ are initialized randomly, which is invertible with probability 1. In addition, because $\det \tilde{W}(u)$ is a continuous function of $u$, it must be nonzero in the neighborhood of the ground truth, which is invertible due to the definition of the symplectic form. Hence, $\tilde{W}$ is expected to be invertible in learning processes and also in prediction.

---

> > ### Comment · Reviewer_XXhP · 2021-08-17
> > **Thanks for the response**
> >
> > Most of my concerns have been addressed. I raised my rating from 6 to 7.

---

> > > ### Author Response · Authors · 2021-08-18
> > > **Thank you very much indeed**
> > >
> > > We are pleased that our response has addressed your concerns. We deeply appreciate the update.

---

### Official Review · Reviewer_qdun · 2021-07-14

**Rating:** 7
**Confidence:** 4

**Summary:**

This paper proposes to learn conservative physical dynamics from data by directly learning a differential 1-form. The dynamics is subsequently determined by computing the exterior derivative of the learned 1-form (yielding a symplectic 2-form) and providing a coordinate-free representation of the dynamics. In addition to inherently preserving the symplectic nature of the dynamics, this procedure does not require *a priori* knowledge of the generalized momentum coordinates, i.e. only velocities need be observed so masses of the degrees of freedom in the data are not needed. The neural symplectic form methodology is compared to an approach in which the symplectic 2-form is learned directly as a skew-symmetric matrix as well the existing Hamiltonian neural network and Lagrangian neural network approaches.

**Limitations And Societal Impact:**

There are a few inherent limitations when it comes to scaling this problem up to large, many-body systems. First, it is hard to imagine a realistic setting in which one could observe all the positions of a complex system with sufficient spatio-temporal resolution. Secondly, most many-body systems will decorrelate over short times due to ergodicity and the onset of chaos. In my mind, this makes the problem a rather academic one without significant applications in physics per se. That said, discovering Hamiltonian structure in ODEs (as in the Lotka-Volterra example) provides a nice counterpoint to this criticism.

**Main Review:**

Learning the underlying Hamiltonian structure of observed trajectorial data requires methodologies that are sensitive to the conservation laws inherent in the physical structure of Hamilton's equations of motion. Without a doubt, conservation of energy is the most important property to preserve. This can be achieved by using a symplectic integration and learning with a symplectic integration scheme noticeably improves the quality of learning (cf. Chen et al. https://arxiv.org/abs/1909.13334 which should surely be cited). Indeed, a direct approach of this type requires rather complete information about the system, including knowledge of all generalized positions and momenta along a time series. This paper aims to relax this requirement by parameterizing a coordinate-free representation of the equations of motion, an elegant idea with significant potential impact.

The formulation of the algorithm relies on (1) representing the Hamiltonian $H$ as a neural network and (2) either representing the symplectic 2-form $W_u$ as a neural network or representing a 1-form as a neural network and computing its exterior derivative to compute $W_u$. These formulations constitute the core contribution of the paper. In both cases, the authors establish formally that the dynamics conserves energy (a comment in the proof of Theorem 1 could be added to indicate that the argument also applies to Theorem 2). The proof of the universal approximation theorem is rather perfunctory, but I do think it constitutes a reasonable addition to this work. The arguments for all three theorems are essentially standard results.

The motivation for preferring the neural symplectic form approach is motivated, in part, by invoking de Rham cohomology. I found this to be a bit of a distraction, likely to lead to confusion. The articulation beginning at line 153 struck me as a much clearer and conceptually better explanation of why one should prefer learning the 1-form. Of course, if this could be clearly tied to the de Rham cohomology, it could be worthwhile to discuss this. Fig. 2 uses a lot of space and did not add meaningfully to the manuscript in my opinion. I appreciated the discussion of the computational cost of the inverse of $W_u$---this is a significant limitation of the algorithm for large systems.

I found the numerical experiments useful, concise, and sufficient to establish the efficacy of the method.

Is learning $W_u$ directly as a 2-form distinct from learning a mass matrix in the case where the kinetic energy is assumed to be harmonic?

It would be useful for the authors to make some contact with the literature on learning potential energy functions (e.g., the work Tkatchenko and coworkers) because in that line of inquiry it has been concluded that learning with forces (learning the gradients of the Hamiltonian) is often superior to learning the potential energy. This strikes me as being analogous to learning the 1-form vs learning the 2-form and I'm not sure how to reconcile the two conclusions.

**Time Spent Reviewing:**

2.5

---

> ### Author Response · Authors · 2021-08-10
> **Response to Reviewer**
>
> Thank you very much for your kind review. We are glad that the reviewer recognized the proposed method as "an elegant idea with significant potential impact" and that "the numerical experiments useful, concise, and sufficient to establish the efficacy of the method." We address your questions as follows.
>
> **Q1: The motivation for preferring the neural symplectic form approach is motivated, in part, by invoking de Rham cohomology. I found this to be a bit of a distraction, likely to lead to confusion. The articulation beginning at line 153 struck me as a much clearer and conceptually better explanation of why one should prefer learning the 1-form. Of course, if this could be clearly tied to the de Rham cohomology, it could be worthwhile to discuss this. Fig. 2 uses a lot of space and did not add meaningfully to the manuscript in my opinion.**
>
> Certainly, the motivation of this study is explained mainly by using the terms of exterior calculus, which may not be accessible for readers.  As you pointed out, the articulation beginning at line 153 "Unlike (3), in this model skew-symmetric matrices that do not correspond to symplectic forms are not explored." is essential. We will reorganize the motivation part of the paper by inserting intuitive phrases like this in the final version. Besides, we will make Fig.2 smaller and add intuitive explanations that do not use the terms of the exterior calculus.
>
> **Q2 I appreciated the discussion of the computational cost of the inverse of Wu---this is a significant limitation of the algorithm for large systems.**
>
> The standard LU decomposition, of which the computational cost is $O(N^3$​) where $N$​ is the size of the matrix, is used in the current implementation; however, perturbation theory of the inverse matrix can be applied. For example, it is known that if the norm of $\Delta M$​ is small enough, $(M+\Delta M)^{-1} \simeq M^{-1}+M^{-1}\Delta M M^{-1}$​ holds. In our case, the skew symmetric matrix depends on the state variable $u$​, which is a continuous function of time $t$​. Hence, the difference between the skew matrices in two successive steps must be small, and hence the perturbation theory can be applied. This reduces the computational costs from $O(N^3$​) to $O(N^2$​). In addition, the matrix multiplication is easily parallelized.
>
> We will add this discussion in the final version.
>
> **Q3: Is learning Wu directly as a 2-form distinct from learning a mass matrix in the case where the kinetic energy is assumed to be harmonic?**
>
> If the kinetic energy is assumed to be harmonic and in addition if the data is given in the coordinate $(q, \dot{q})$​​, the skew symmetric matrix can be obtained using the mass matrix. Suppose that the kinetic energy is given as $\frac{1}{2}\dot{q}^\top M \dot{q}$​​ and the potential energy as $V(q)$​​. Then, with the Hamiltonian $H = \frac{1}{2}\dot{q}^\top M \dot{q} + V(q)$​​ we get
> $$
> \frac{\mathrm{d}}{\mathrm{d}t}\pmatrix{q \\\\ \dot{q}}
> = \\pmatrix{\dot{q} \\\\ - M^{-1} \frac{\partial V}{\partial q}}
> =\\pmatrix{O & M^{-1} \\\\ -M^{-1} & O} \\pmatrix{\frac{\partial V}{\partial q} \\\\ M \dot{q}}
> =\\pmatrix{O & M^{-1} \\\\ -M^{-1} & O} \\pmatrix{\frac{\partial H}{\partial q} \\\\ \frac{\partial H}{\partial \dot{q}}}.
> $$
> Hence, the symplectic form is given by the inverse of $\\pmatrix{O & M^{-1} \\\\ -M^{-1} & O}$.
> We emphasize that the advantage of the proposed neural symplectic form is that it is applicable to any other coordinates or a non-harmonic kinetic energy, unlike learning a mass matrix.
>
> **Q4: It would be useful for the authors to make some contact with the literature on learning potential energy functions (e.g., the work Tkatchenko and coworkers) because in that line of inquiry it has been concluded that learning with forces (learning the gradients of the Hamiltonian) is often superior to learning the potential energy. This strikes me as being analogous to learning the 1-form vs learning the 2-form and I'm not sure how to reconcile the two conclusions.**
>
> Many thanks for the reference. We have read some papers on this subject. Since the energy function and its derivative are considered to be a 0-form and a 1-form, there may be a connection with this study.  Their research seems to be mainly on chemistry, which we are not very familiar with, unfortunately. We will investigate the connection in future studies.
>
> We will cite the work by Tkatchenko and coworkers and also Chen et al. https://arxiv.org/abs/1909.13334 in the final version.

---

### Official Review · Reviewer_yYL7 · 2021-07-16

**Rating:** 6
**Confidence:** 4

**Summary:**

Authors present a ML-based approach to learning dynamics in the space of Hamiltonian systems. The approach is based on generic formulation of Hamiltonian mechanics in the language of differential geometry, effectively avoiding the requirement of knowing the appropriate generalized coordinates beforehand. The proposed approach is based on modeling the symplectic 2-form (known to be closed) as an exterior derivative of a parametrized (learned) 1-form. This method is termed "neural symplectic form" and provides a compact parametrization of the underlying governing equations.

The proposed model is trained by minimizing the MSE between estimated ground truth time derivatives of the dynamical system and predictions of the model on several datasets: mass-spring system; double pendulum; Lotka-Volterra equations (which are not captured by a traditional formulation of Hamiltonian mechanics with fixed symplectic form).

The model is compared to baselines (HNN, LNN and a naive implementation termed skew matrix learning) on test loss and errors of the conserved quantity in the predicted dynamics accumulated during simulation for a fixed time.

Authors find that the proposed model produces best results across most of the comparisons, especially on the Lotka-Volterra test case.



**Limitations And Societal Impact:**

The work does not have negative societal impact.

**Main Review:**

The article present an original approach to modeling Hamiltonian systems that uses fewer variational parameters while accurately capturing the target space of dynamical laws. When combined with symplectic integrators, such systems admit conservation laws that are highly desirable for long-term modeling. In my opinion this is a significant direction worthy further exploration.

The work is written clearly, but is definitely more accessible to readers with elementary background in exterior calculus. One of the organization caveats that in my opinion should be addressed is the silent push of the NODE model to the appendix, which often perform on-par with the proposed method.

The results section raises a few questions:
1) Some of the error-bars are comparable/larger than the mean values? Is it possible to include more experiments to make comparisons precise?
2) While summarized errors are useful it would be valueable to include the time dependence of the errors as a function of time (for both energy and MAE/MSE)
3) The networks used for modeling the system were MLP with a single hidden layer - it would be very reassuring to see that the results are not too sensitive to this choice, especially in a more expressive regime.

Overall - interesting paper worth publishing assuming that the results section can be tuned up.

**Time Spent Reviewing:**

5

---

> ### Author Response · Authors · 2021-08-10
> **Response to Reviewer**
>
> Thank you very much for your kind and detailed reviewing.
>
> **Q1: The work is written clearly, but is definitely more accessible to readers with elementary background in exterior calculus. One of the organization caveats that in my opinion should be addressed is the silent push of the NODE model to the appendix, which often perform on-par with the proposed method.**
>
> For the part that uses exterior calculus, we will try to make the text easier to understand in the final version. Regarding the NODE model, we pushed the experiments of NODE models to the appendix because we suppose that the usefulness of the Hamiltonian structure for long-term predictions has been established in the previous studies; however, as you suggested, for better interpretability we will include the comparisons with the NODE models to the main part in the final version, replacing the results in the submitted paper with the those shown below.
>
> **Q2: The networks used for modeling the system were MLP with a single hidden layer - it would be very reassuring to see that the results are not too sensitive to this choice, especially in a more expressive regime.**
>
> Thank you very much for pointing this out. First of all, the "single hidden-layer" mentioned in the submitted paper is actually a typo: the networks have two hidden-layers each of which has 50 units. This can be confirmed by taking a look at the codes in Supplemental Materials.
>
> Besides, we have performed additional experiments in the standard setting of HNNs, in which the networks have two hidden-layers with 200 units and the number of training epochs is 2000. In addition, to reduce the effects of rounding error, we performed the experiments with double precision using Titan V GPUs. As summarized in the table below, the tendency of the results is almost the same as the one shown in the submitted paper. Hence, we will leave the discussions as they are and replace the results with the ones below in the final version.
>
> **test time-derivative errors**
>
> |                     |   skew matrix learning    |    neural symplectic form     |             NODE              |            HNN            |         LNN         |          LNN(best)          |
> |:--------------------|:-------------------------:|:-----------------------------:|:-----------------------------:|:-------------------------:|:-------------------:|:---------------------------:|
> | **mass spring**     | 9.614e-05 $\pm$ 8.525e-05 | **1.214e-07** $\pm$ 5.208e-08 |   3.912e-07 $\pm$ 3.384e-07   | 7.923e-05 $\pm$ 6.286e-07 | 7.786 $\pm$ 23.357  | *1.381e-07* $\pm$ 1.050E-07 |
> | **double pendulum** | 3.775e-04 $\pm$ 2.964e-04 | **3.909e-05** $\pm$ 9.704e-06 |  *1.882e-04* $\pm$ 2.555e-05  | 3.821e-03 $\pm$ 5.100e-04 | 23.210 $\pm$ 36.156 |   2.092 e-2 $\pm$ 1.116e-4   |
> | **Lotka-Volterra**  | 6.916e-07 $\pm$ 1.697e-07 |  *1.917e-07* $\pm$ 1.264e-07  | **2.976e-11** $\pm$ 2.041e-11 | 1.406e-04 $\pm$ 2.913e-06 |         N/A         |             N/A             |
>
> We have also performed two other experiments. Firstly, we have investigated long-term behaviors of the models for the Lotka-Volterra test. Please note that those for the double pendulum test are included in Supplemental Materials (Figure 11.) Although we cannot show the resulting images due to the limitation of OpenReview, the prediction by the NODE models failed in the sense that the height of the peaks gradually increase or decrease, while the proposed model did not. As a quantitative evaluation, the error of the true energy function $H$ at $t=100$, $H(u(100))-H(u(0))$, was 50.672 $\pm$ 79.431 for the NODE model and 6.841e-02 $\pm$ 7.448e-02 for the proposed model. The NODE model has the large error-bar because the energy can increase or decrease. We will include these results in the final version.
>
> Secondly, we have performed a test of image-based dynamics learning. We performed the Pixel test of the HNN's code. Please note that, in this test, HNN is combined with an autoencoder and the whole model is trained by minimizing the loss function
> $$
> loss_{AE} + loss_{CC}+0.1\ loss_{HNN}
> $$
> in which not only the loss functions for the autoencoder ($loss_{AE}$) and the HNN ($loss_{HNN}$), but also the canonical coordinate loss ($loss_{CC}$) is included. As you can see in the comment of the code (https://github.com/greydanus/hamiltonian-nn/blob/master/experiment-pixels/train.py#L57), the canonical coordinate loss is introduced to "make(s) latent space look like (x, v)." Hence, essentially, an analytic expression $p = f(q, \dot{q})$ of the generalized momentum $p$ is assumed to be known in this code, and the canonical coordinate loss defined by $\| p - f(q, \dot{q}) \|$ is needed for good learning.
>
> The proposed neural symplectic form does not require the analytic expression of the generalized momentum. Therefore, we can learn the dynamics without knowing the analytical expression, in other words, we can learn the dynamics by minimizing only the other two terms:
> $$
> loss_{AE} + 0.1\ loss_{HNN}.
> $$
> Since images are not supported in OpenReview, we cannot show the results here, but the proposed neural symplectic form gives almost the same results as the HNN with the canonical coordinate loss. In addition, the average $L^1$ error between the ground-truth images and the predicted images by the proposed method over 1000 frames was 36.582, which is smaller than that of HNN, 40.746.
>
> We will include these experiments in the final version along with the resulting images.
>
> **The results section raises a few questions:**
>
> **Q3: Some of the error-bars are comparable/larger than the mean values? Is it possible to include more experiments to make comparisons precise?**
>
> Given the new results in Q2, almost all the error-bars are now smaller than the mean values. Only the exception is those for LNNs. As discussed in Appendix D, because the Lagrangian is not uniquely determined, the learning process of LNNs was unstable (see also Figure 8 in Supplemental Materials), resulting in the large error-bars. We took the best snapshots of LNNs and summarized their results, whose error-bars are small enough (see the column named LNNs (best)).
>
> **Q4: While summarized errors are useful it would be valueable to include the time dependence of the errors as a function of time (for both energy and MAE/MSE)**
>
> Thank you very much for the suggestion. We will include the time dependence of the errors as a function of time in the final version.

---

> > ### Comment · Reviewer_yYL7 · 2021-08-31
> > **Comment**
> >
> > Thank you for providing the updated results and comments on the review. I have a few questions that I'd like to follow up on:
> >
> > >  The NODE model has the large error-bar because the energy can increase or decrease.
> >
> > Since we are interested in the absolute energy error, which is a positive-definite quantity, the error bars can be reduced by increasing the number of samples regardless whether the underlying error can be positive, negative or both?
> >
> > > We will include the time dependence of the errors as a function of time in the final version.
> >
> > It would be very helpful to have such assessment as a primary way of model comparison (rather than comparing the predicted time derivatives). While OpenReview does not support pasting images, it is permitted (to my knowledge) to provide supporting images on anonymized hosting services, which in this case would be quite helpful.
> >
> > Thank you

---

> > > ### Author Response · Authors · 2021-09-02
> > > **Thank you very much for your comments.**
> > >
> > > Thank you very much for your comments.
> > >
> > > > Since we are interested in the absolute energy error, which is a positive-definite quantity, the error bars can be reduced by increasing the number of samples regardless whether the underlying error can be positive, negative or both?
> > >
> > > We are sorry for the confusing explanation. We meant that NODE often gets a large error by nature because NODE is not guaranteed to conserve the energy, implying that its orbit can diverge and its error bar can diverge as well.
> > >
> > > > > We will include the time dependence of the errors as a function of time in the final version.
> > > >
> > > > It would be very helpful to have such assessment as a primary way of model comparison (rather than comparing the predicted time derivatives). While OpenReview does not support pasting images, it is permitted (to my knowledge) to provide supporting images on anonymized hosting services, which in this case would be quite helpful.
> > >
> > > Thank you very much for your suggestion; we had not thought of using anonymized hosting services. We uploaded the images that show the typical time dependences of the errors. Please see https://ibb.co/7yF130s. In all tests, the solutions of the target system exhibit quasi-periodic behaviors, and hence the errors show periodicity to a certain extent. In addition, the double pendulum is difficult to predict and the errors are larger for all methods than the other tests because of its chaotic nature. As mentioned above, due to the non-existence of the energy conservation law, the errors in the energy and state for NODE often diverge. The errors of the proposed neural symplectic form (denoted as Symp) are confirmed to be smaller than or at least similar to the others.

---

> > > > ### Comment · Reviewer_yYL7 · 2021-09-02
> > > > **Comment**
> > > >
> > > > > Please see https://ibb.co/7yF130s.
> > > >
> > > > Is the y-axis scaled misrepresented? Looking at the double pendulum results the MAE errors are shown much larger than expected. Even eyeballing Fig. 5 from the original paper would suggest that the errors should not be on the order 1 for such short time scales. Is there a global axis scale missing?

---

> > > > > ### Author Response · Authors · 2021-09-02
> > > > > **Thank you very much for your comments again**
> > > > >
> > > > > Thank you very much for your careful review, and sorry for confusing notions.
> > > > >
> > > > > We normalized all data to facilitate neural network training. We reported the squared errors (around $10^{-4}$ for double pendulum) in normalized states in Tables. The normalizing scale is around $10^{-2}$ ( for double pendulum, contained in the provided codes). Therefore, the expected absolute errors in the unnormalized states are around $\sqrt{10^{-4}}/10^{-2}=10^0$.
> > > > >
> > > > > We reported images of the absolute errors (around $10^0$ per unit time for double pendulum) of unnormalized states, which is consistent with the errors expected above.
> > > > >
> > > > > To eliminate this confusion, we will provide all results in a consistent scale in the final version.

---

> > > > > > ### Comment · Reviewer_yYL7 · 2021-09-02
> > > > > > **Reply**
> > > > > >
> > > > > > Thanks for clarifying and the hard work providing additional details and plots!
> > > > > >
> > > > > > Given that the double pendulum is a chaotic system it might be interesting to show the onset of such chaotic behavior by lookin at averages MAE/MSE on a larger time scale. This might be an easy and pedagogical addition for the final version (probably an appendix material).
> > > > > >
> > > > > > Thanks again for providing extra details and prompt responses - I've raised by score to 6.

---

> > > > > > > ### Author Response · Authors · 2021-09-02
> > > > > > > **Thank you very much indeed**
> > > > > > >
> > > > > > > Many thanks for your kind review and suggestions, which are really helpful for improving the paper. We will add the suggested investigation in the final version. Thank you very much indeed again.

---

### Official Review · Reviewer_MBYz · 2021-07-16

**Rating:** 7
**Confidence:** 3

**Summary:**

This work addresses the problem of learning Hamiltonian systems from data without assuming that the generalized momentum is given. This assumption limits the applicability of standard Hamiltonian Neural Networks (HNNs) because the generalized momentum is defined in terms of the Hamiltonian itself, which means it cannot typically be provided as training data for unknown Hamiltonians. This work proposes to get around this limitation by exploiting a coordinate-free formulation of the Hamiltonian and incorporating the 2-form from symplectic geometry into the structure of a HNN (either directly as a learned skew-symmetric matrix or via a learned 1-form and an exterior derivative). The resulting architecture performs favorably compared to a standard HNN and Lagrangian Neural Network (LNN) on several simple physical systems.

**Ethical Concerns:**

None.

**Limitations And Societal Impact:**

Yes.

**Main Review:**

Positives:
- By eliminating the requirement for labeled generalized momenta at input, the core technical contribution of the paper overcomes a key limitation of HNNs and does so in a simple and elegant way.
- The paper does an admirable job exposing the relevant literature in a way that I feel will be largely interpretable by the NeurIPS audience, despite the fact that it draws heavily from symplectic geometry. In particular, I found the discussion of 1- and 2-forms and the exterior derivative very readable.

Negatives:
- My largest concern with this paper lies in the experimental sections. Experiments were conducted with very small (single hidden-layer) networks, which are smaller than typically used in this literature. This appears to be leading to artificially bad performance by HNN and LNN baselines, as discussed in L269-272 and section D. While it's suggestive that the proposed method works well with this network, it makes the results very hard to interpret in the context of the literature. This choice isn't motivated as far as I could tell. For the method to be interpretable, it should be run in settings where HNNs and LNNs are known to perform well.
- The model state prediction should be analyzed quantitatively: how well does the model predict the state when integrated over time? It's hard to interpret the quality of the estimates without knowing how it fares over longer horizons than a single step, which is essentially what's quantified in tables 2 and 3.
- How were the energy errors in Table 3 estimated? As I understand it, these models can only recover the energy up to a constant offset.
- I believe the paper could be much more impactful if more difficult experimental settings where the proposed method can shine were explored. In particular, the proposed formulation appears to be particularly amenable to image- or other observation-based depictions of Hamiltonian systems, because the method does not depend on a representation being recovered in the Darboux coordinate system and does not require momentum to be estimated. Image-based Hamiltonian learning has been addressed in several prior works in the literature e.g. [1, 2, 3], and it would be very useful to know how the proposed method fares in those settings and how the present analysis sheds light on those experiments.
- Several of the figures could be improved:
  - Figure 1 is cluttered and hard to follow: this could probably be improved by moving many of the graphical elements to the text description, which should be expanded. I also found it hard to read the left-hand side of the figure, as the double pendulum and Lotka-Volterra systems look as if they're both being input to the system (rather than showing two example inputs).
  - Figure 2 is very hard to follow and is essentially uninterpretable without referring to the main text.
  - Figures 5, 6, and 7 are missing axis labels and do not refer to what system they evaluate.

Minor:
- L92: extra "the" at the beginning of the sentence
- L115: (4) should be (2).
- I found it a little hard to understand the structure Y_{NN} on a first read. I believe this would be easier to understand if its input and output dimensions were mentioned.
- L190: "If the time derivatives are not available, interpolated data should be used." Please rephrase this - I believe the implication is that finite-difference estimates of the temporal derivatives should be used.
- L204: Nitpick: I believe a more idiomatic way to refer to this section in a machine learning paper would be "Experiments" or "Numerical Experiments".

Overall: I believe this work makes a solid contribution to the literature on learning Hamiltonians and the weaknesses are addressable, so I weakly recommend acceptance. However, I think the paper would be much stronger if the experimental section were improved.

I am willing to raise my score if the experimental limitations pointed above are addressed, and ideally if the authors can present results on more challenging datasets.

[1] Greydanus et al - NeurIPS 2019 - Hamiltonian Neural Networks
[2] Toth et al - ICLR 2020 - Hamiltonian Generative Networks
[3] Allen-Blanchette et al - AAAI 2020 - LagNetViP: A Lagrangian Neural Network for Video Prediction

**Time Spent Reviewing:**

3

---

> ### Author Response · Authors · 2021-08-10
> **Response to Reviewer**
>
>   **Q1: My largest concern with this paper lies in the experimental sections. Experiments were conducted with very small (single hidden-layer) networks, which are smaller than typically used in this literature. This appears to be leading to artificially bad performance by HNN and LNN baselines, as discussed in L269-272 and section D. While it's suggestive that the proposed method works well with this network, it makes the results very hard to interpret in the context of the literature. This choice isn't motivated as far as I could tell. For the method to be interpretable, it should be run in settings where HNNs and LNNs are known to perform well.**
>
>   Thank you very much indeed for your detailed reviewing. First of all, the "single hidden-layer" mentioned in the submitted paper is actually a typo: the networks have two hidden-layers each of which has 50 units. This can be confirmed by taking a look at the codes in Supplemental Materials.
>
>   Besides, we have performed additional experiments in the standard setting of HNNs, in which the networks have two hidden-layers with 200 units and the number of training epochs is 2000. In addition, to reduce the effects of rounding error, we performed the experiments with double precision using Titan V GPUs. As summarized in the table below, the tendency of the results is almost the same as the one shown in the submitted paper. Hence, we will leave the discussions as they are and replace the results with the ones below in the final version.
>
> **test time-derivative errors**
>
> |                     |   skew matrix learning    |    neural symplectic form     |             NODE              |            HNN            |         LNN         |          LNN(best)          |
> |:--------------------|:-------------------------:|:-----------------------------:|:-----------------------------:|:-------------------------:|:-------------------:|:---------------------------:|
> | **mass spring**     | 9.614e-05 $\pm$ 8.525e-05 | **1.214e-07** $\pm$ 5.208e-08 |   3.912e-07 $\pm$ 3.384e-07   | 7.923e-05 $\pm$ 6.286e-07 | 7.786 $\pm$ 23.357  | *1.381e-07* $\pm$ 1.050E-07 |
> | **double pendulum** | 3.775e-04 $\pm$ 2.964e-04 | **3.909e-05** $\pm$ 9.704e-06 |  *1.882e-04* $\pm$ 2.555e-05  | 3.821e-03 $\pm$ 5.100e-04 | 23.210 $\pm$ 36.156 |   2.092 e-2 $\pm$ 1.116e-4   |
> | **Lotka-Volterra**  | 6.916e-07 $\pm$ 1.697e-07 |  *1.917e-07* $\pm$ 1.264e-07  | **2.976e-11** $\pm$ 2.041e-11 | 1.406e-04 $\pm$ 2.913e-06 |         N/A         |             N/A             |
>
> Note that NODE failed in a longer-time prediction (see Figure 11 in Supplemental Materials).
>
> **Q2: The model state prediction should be analyzed quantitatively: how well does the model predict the state when integrated over time? It's hard to interpret the quality of the estimates without knowing how it fares over longer horizons than a single step, which is essentially what's quantified in tables 2 and 3.**
>
> Sorry for the confusion, but we had investigated long-term behaviors for the double pendulum test in Supplemental Materials (see Figure 11 for example). The predicted state by NODE vanished or diverged, while the proposed model sustained the oscillation. Moreover, we newly investigated long-term behaviors for the Lotka-Volterra test. Similarly to the double pendulum case, the prediction by NODE failed in the sense that the height of the peaks gradually increase or decrease, while the proposed model sustained the peak height of the periodic oscillation. As a quantitative evaluation, for the Lotka-Volterra test, the error of the true energy function $H$ at $t=100$, $H(u(100))-H(u(0))$, was 50.672 $\pm$ 79.431 for the NODE model and 6.841e-02 $\pm$ 7.448e-02 for the proposed model. The NODE model has the large error-bar because the energy can increase or decrease. We will include these results into the main part of the paper in the final version.
>
> **Q3: How were the energy errors in Table 3 estimated? As I understand it, these models can only recover the energy up to a constant offset.**
>
> Yes, you are right. Following the previous work, we normalized the energy so that it becomes 0 at $t=0$. In other words, as described in the caption of Table 3, the energy errors are measured by $H(u(T)) - H(u(0))$, where $H$ is the true Hamiltonian and $T=30$ for the Lotka-Volterra test and $T=5$ for the other two in the experiments in Section 4.
>
> **Q4: I believe the paper could be much more impactful if more difficult experimental settings where the proposed method can shine were explored. In particular, the proposed formulation appears to be particularly amenable to image- or other observation-based depictions of Hamiltonian systems, because the method does not depend on a representation being recovered in the Darboux coordinate system and does not require momentum to be estimated. Image-based Hamiltonian learning has been addressed in several prior works in the literature e.g. [1, 2, 3], and it would be very useful to know how the proposed method fares in those settings and how the present analysis sheds light on those experiments.**
>
>   Many thanks for this suggestion. We agree with you and applied the proposed neural symplectic form to an image-based Hamiltonian learning. We performed the Pixel test of the HNN's code. Please note that, in this test, HNN is combined with an autoencoder and the whole model is trained by minimizing the loss function
>   $$
>   loss_{AE} + loss_{CC}+0.1\ loss_{HNN}.
>   $$
>   In addition to the loss functions for the autoencoder ($loss_{AE}$​) and the HNN ($loss_{HNN}$​), the canonical coordinate loss ($loss_{CC}$​) is included. As you can see in the comment of the code (https://github.com/greydanus/hamiltonian-nn/blob/master/experiment-pixels/train.py#L57), the canonical coordinate loss is introduced to "make(s) latent space look like (x, v)." Hence, essentially, an analytic expression $p = f(q, \dot{q})$​ of the generalized momentum $p$​ is assumed to be known in this code, and the canonical coordinate loss defined by $\| p - f(q, \dot{q}) \|$​ is needed for good learning.
>
>   The proposed neural symplectic form does not require the analytic expression of the generalized momentum. Therefore, we can learn the dynamics without knowing the analytical expression, in other words, we can learn the dynamics by minimizing only the other two terms:
>   $$
>   loss_{AE} + 0.1\ loss_{HNN}.
>   $$
>   Since images are not supported in Openreview, we cannot show the results here, but the proposed neural symplectic form gives almost the same results as the HNN with the canonical coordinate loss. In addition, the average $L^1$ error between the ground-truth images and the predicted images by the proposed method over 1000 frames was 36.582, which is smaller than that of HNN, 40.746.
>
> We will include these experiments in the final version along with the resulting images.
>
> **Q5: Several of the figures could be improved:**
>
> - **Figure 1 is cluttered and hard to follow: this could probably be improved by moving many of the graphical elements to the text description, which should be expanded. I also found it hard to read the left-hand side of the figure, as the double pendulum and Lotka-Volterra systems look as if they're both being input to the system (rather than showing two example inputs).**
> - **Figure 2 is very hard to follow and is essentially uninterpretable without referring to the main text.**
> - **Figures 5, 6, and 7 are missing axis labels and do not refer to what system they evaluate.**
>
> **Minor things:**
>
> Thank you very much for pointing these out. We will improve or correct these in the final version.

---

> > ### Comment · Reviewer_MBYz · 2021-09-01
> > **Concern about change in results**
> >
> > Dear authors - thank you for your detailed response. Although the new results described in your response are much stronger, I'm concerned that the systematic improvement now makes it difficult to evaluate the original submission. The new results improve on the submitted results for every method in the paper by several order of magnitude. In my opinion, this means that every figure and table in the paper will need to be updated. I feel the paper would be easier to evaluate if as many figures and tables as possible were updated and linked anonymously. Otherwise, I feel that it is difficult to evaluate the revised paper.
> >
> > The new results also require a qualitative change in the interpretation of the results: e.g. LNN was reported to diverge in the original submission but it no longer does. The Appendix of the original submission includes a page of analysis of why LNN diverges, but the new results now suggest the instability may be in large part due to poor tuning. Can you please comment on this point?
> >
> > Unfortunately, these new results also lead me to be somewhat suspicious about the results. The original paper misreported the network configuration. The newly reported results are dramatically better (by several orders of magnitude in all cases). Were these the only two numbers reported? What hyperparameter search procedure was used to choose these two configurations?

---

> > > ### Author Response · Authors · 2021-09-01
> > > **Thank you very much for your comment.**
> > >
> > > Thank you very much for your comment.
> > >
> > > > Although the new results described in your response are much stronger, I'm concerned that the systematic improvement now makes it difficult to evaluate the original submission. The new results improve on the submitted results for every method in the paper by several order of magnitude.
> > >
> > > Sorry for confusing notions, but the new results are only slightly better than the original results. In Table 2 of the original submission, we omitted the scale of $10^{-6}$​​​ from Lotka-Volterra equation and the scale of $10^{-4}$​​​ from others to improve the visibility (please see the marginal remarks of Table 2.) For convenience, the two results are presented again below for comparison. For example, for double pendulum, the neural symplectic form got the error of $0.55\times 10^{-4}=5.5\times 10^{-5}$​​​ in the original results and the error of $3.9\times 10^{-5}$​​​​ in the new results. Therefore, all discussions about the performances in the original submission still hold. In particular, with the new experimental settings, LNN can still diverge, especially for double pendulum, for which the error is still very large compared to the other methods.
> > >
> > > We are also working on replacing figures and tables in the manuscript. We will upload it when we finish replacing them.
> > >
> > > **test time-derivative errors (the results in the original paper)**
> > >
> > > |                     |  skew matrix learning   |   neural symplectic form    |            NODE             |           HNN           |           LNN            |        LNN(best)        |
> > > | :------------------ | :---------------------: | :-------------------------: | :-------------------------: | :---------------------: | :----------------------: | :---------------------: |
> > > | **mass spring**     | 1.64e-04 $\pm$ 1.03e-04 | **2.76e-07** $\pm$​​ 2.14e-07 |  *1.94e-06* $\pm$​ 2.56e-06  | 8.41e-05 $\pm$​​ 9.23e-07 |    42.74 $\pm$ 128.22    | 3.68e-06 $\pm$​​ 4.34e-06 |
> > > | **double pendulum** | 3.05e-04 $\pm$ 5.08e-05 | **5.49e-05** $\pm$ 1.01e-05 |  *2.14e-04* $\pm$ 0.19e-04  | 3.54e-03 $\pm$ 4.52e-05 | 81138.12 $\pm$ 176150.94 |     1.49 $\pm$​ 4.41     |
> > > | **Lotka-Volterra**  | 7.06e-06 $\pm$​ 7.88e-06 |  *4.64e-07* $\pm$​ 2.18e-07  | **5.06e-11** $\pm$ 4.26e-11 | 1.42e-04 $\pm$ 1.82e-06 |           N/A            |           N/A           |
> > >
> > > **test time-derivative errors (the results in the previous response)**
> > >
> > > |                     |   skew matrix learning    |    neural symplectic form     |             NODE              |            HNN            |         LNN         |          LNN(best)          |
> > > | :------------------ | :-----------------------: | :---------------------------: | :---------------------------: | :-----------------------: | :-----------------: | :-------------------------: |
> > > | **mass spring**     | 9.614e-05 $\pm$ 8.525e-05 | **1.214e-07** $\pm$ 5.208e-08 |   3.912e-07 $\pm$ 3.384e-07   | 7.923e-05 $\pm$ 6.286e-07 | 7.786 $\pm$ 23.357  | *1.381e-07* $\pm$ 1.050e-07 |
> > > | **double pendulum** | 3.775e-04 $\pm$ 2.964e-04 | **3.909e-05** $\pm$ 9.704e-06 |  *1.882e-04* $\pm$ 2.555e-05  | 3.821e-03 $\pm$ 5.100e-04 | 23.210 $\pm$ 36.156 |   2.092e-2 $\pm$ 1.116e-4   |
> > > | **Lotka-Volterra**  | 6.916e-07 $\pm$ 1.697e-07 |  *1.917e-07* $\pm$ 1.264e-07  | **2.976e-11** $\pm$ 2.041e-11 | 1.406e-04 $\pm$ 2.913e-06 |         N/A         |             N/A             |

---

> > > > ### Comment · Reviewer_MBYz · 2021-09-01
> > > > **Thank you for clarifying**
> > > >
> > > > Thanks to the authors for their prompt and clear response: I indeed misinterpreted the relationship between the old and new numbers. Given the relatively minor change in the results, I no longer believe it's necessary to see new plots to make a clear evaluation. The authors have answered my other questions satisfactorily and I will raise my score to a 7.

---

> > > > > ### Author Response · Authors · 2021-09-02
> > > > > **Thank you very much indeed**
> > > > >
> > > > > Thank you very much indeed for your response and the update. We are pleased to hear that your concerns have been addressed.

---

### Decision · Program_Chairs · 2021-09-27

**Decision:**

Accept (Spotlight)

**Comment:**

All ratings were "accept". The reviewers raised various suggestions which the authors responded to thoroughly, with additional experiments that reinforced their results. This should be a solid contribution.